# TAILORING SELF-RATIONALIZERS WITH MULTI-REWARD DISTILLATION

**Sahana Ramnath**[♥]**, Brihi Joshi**[♥]**, Skyler Hallinan**[♣]**, Ximing Lu**[♣]**, Liunian Harold Li**[♠]
**Aaron Chan**[♥]**, Jack Hessel**[◇]**, Yejin Choi**[♣◇]**, Xiang Ren**[♥◇]

[♥]University of Southern California [♣]University of Washington
[♠]University of California Los Angeles [◇]Allen Institute for Artificial Intelligence
sramnath@usc.edu

## ABSTRACT

Large language models (LMs) are capable of generating *free-text rationales* to aid question answering. However, prior work 1) suggests that useful self-rationalization is emergent only at significant scales (e.g., 175B parameter GPT-3); and 2) focuses largely on downstream performance, ignoring the semantics of the rationales themselves, e.g., are they faithful, true, and helpful for humans? In this work, we enable small-scale LMs (~200x smaller than GPT-3) to generate rationales that not only improve downstream task performance, but are also more plausible, consistent, and diverse, assessed both by automatic and human evaluation. Our method, MARIO (**M**ulti-rew**A**rd **R**at**IO**nalization), is a multi-reward conditioned self-rationalization algorithm that optimizes multiple distinct properties like plausibility, diversity and consistency. Results on five difficult question-answering datasets StrategyQA, QuaRel, OpenBookQA, NumerSense and QASC show that not only does MARIO improve task accuracy, but it also improves the self-rationalization quality of small LMs across the aforementioned axes better than a supervised fine-tuning (SFT) baseline. Extensive human evaluations confirm that MARIO rationales are preferred vs. SFT rationales, as well as qualitative improvements in plausibility and consistency[1].

## 1 INTRODUCTION

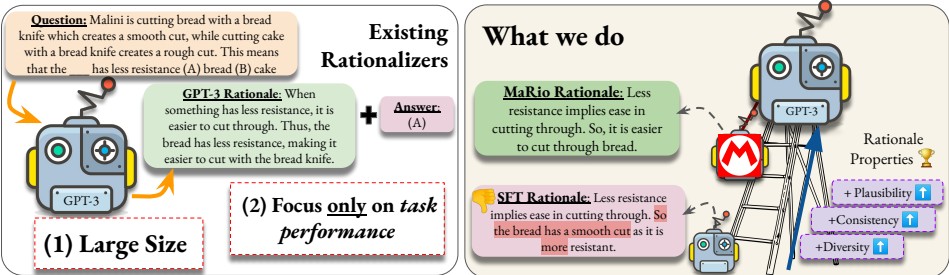

Figure 1: **Our proposed approach, MARIO.** While existing self-rationalizing pipelines require exorbitantly large LMs that are used to primarily improve task performance, MARIO is a small LM that is initially distilled from rationales generated by GPT-3, following by multi-reward training that improves its rationale quality w.r.t three properties: plausibility, diversity and consistency.

In recent years, there has been a surge of interest in *self-rationalizing* LMs, LMs that generate fluent, human-like, *free-text rationales* that can explain their decision (Wiegreffe et al., 2021). Early approaches in self-rationalizing involved collecting human-written gold rationales and using them as supervision for training LMs (Wiegreffe et al., 2021; Narang et al., 2020). Now, with the advent

---

[1]inklab.usc.edu/MaRio/

of large LMs, chain-of-thought prompting (Wei et al., 2022; Marasovic et al., 2022; Kojima et al., 2022) has revolutionized the landscape of self-rationalization; now, with just a few well-designed prompts and demonstrations, LMs can generate explanations for their predictions. The presence of rationales can make LMs both more interpretable (Lertvittayakumjorn & Toni, 2021) and more usable (Joshi et al., 2023) from the perspective of users.

However, prior work in self-rationalization has largely overlooked the *quality* of generated rationales themselves; instead, their utility is justified by measuring downstream task performance (Wei et al., 2022; Zelikman et al., 2022). This is particularly problematic, as downstream models and users may use these rationales as justifications for the predicted answer, which can further propagate these negative quality outcomes (Atanasova et al., 2023; Joshi et al., 2023; Hovy & Prabhumoye, 2021). Furthermore, it is observed that rationalization comparable to human quality is only observed with at a significant LM parameter scales (~100B or more) (Wei et al., 2022). Despite some recent interest in using smaller LMs for rationalization (Chen et al., 2023b), it is still unclear if smaller LMs can be used to generate similarly high-quality rationales.

In this work, we propose MARIO, a method that focuses on tailoring small-sized LMs (< 1B parameters) to be strong rationalizers both in terms of improved downstream performance, and in terms of desireable properties of the rationales themselves. Instead of relying on human rationale labelling (Wiegreffe et al., 2021), MARIO considers a setting where a small LM only has access to *rewards* that measures factors underlying rationale quality, e.g. a trained LM that judges the plausibility of a rationale and provides a numerical score. MARIO first starts with training a small LM to self-rationalize, with the help of GPT-3 (Brown et al., 2020) (TEXT-DAVINCI-003)[2] generated rationales as initial supervision, which are shown to be of higher quality Sun et al. (2022). It then casts the problem into a multi-reward conditioned rationale generation problem, where the LM is optimized to maximize quality rewards. In order to achieve this, MARIO extends QUARK proposed by Lu et al. (2022) to a multi-reward setup, where generations from an LM are binned according reward values; the LM learns distributions conditioned on 'control-tokens' corresponding to every reward and *high-quality* generations can be obtained via conditioning on the highest-reward token.

We determine that *high-quality* rationales should have three necessary properties: *plausibility* (makes logical sense), *diversity* (is not repetitive) and *consistency* (supports the correct answer for the instance). Generated rationales' rewards are assessed through automated metrics for each of the three quality properties. We then evaluate MARIO on three question-answering datasets, and observe that small LMs like T5-LARGE can be effectively trained to generate rationales that satisfy all of the quality requirements, while *also* leading to improvements in task performance over supervised fine-tuned self-rationalizers (SFT). Via human evaluation, we also observe that rationales generated by MARIO are more preferred over those generated by SFT, across all datasets.

We note that tailoring small LMs with multiple quality rewards is a challenging task. Some of these issues include finding high-quality, stable rewards that can be effectively incorporated in a self-rationalizing pipeline. We also observed that a lot of additional desirable properties in rationales (like factuality and completeness) do not have reliable automated rewards. Furthermore, improving task accuracy (which is the primary goal while generating rationales for a lot of these tasks) is challenging in a multi-reward setup, and show that adding task accuracy as an additional reward term leads to the best configuration of MARIO. By using small LMs to generate high-quality rationales that are also supported by human evaluations, we believe our findings can help guide future work in efficient, real-world situated methods in rationale generation and evaluation.

## 2 SELF-RATIONALIZATION

Throughout this work, we refer to *self-rationalizers* as LMs that are trained or prompted to specifically generate free-text rationales, along with their predictions. These free-text rationales are treated as explanations for their predictions. For the purpose of our experiments, we explore self-rationalization on the question answering (QA) task. Specifically, given a question, the LM must first generate a free-text rationale that explains the LM's reasoning process, followed by an answer to the given question. Table 1 shows examples of inputs and outputs by these self-rationalizing LMs for five QA datasets: STRATEGYQA (Geva et al., 2021), QUAREL (Tafjord et al., 2019), OPEN-

---

[2] note that whenever we mention GPT-3 in this work, we are referring to TEXT-DAVINCI-003

Table 1: **Sample Inputs and Outputs for Self-Rationalizing LMs.** We use an **I-RO** setting for all our experiments. This table shows one example each from the training set of STRATEGYQA, OPENBOOKQA, QUAREL, NUMERSENSE and QASC. The rationales shown here are the ones sampled from GPT-3.

| | |
|---|---|
| STRATEGYQA | INPUT (I): Could someone in Tokyo take a taxi to the The Metropolitan Museum of Art? 
 OUTPUT (RO): The Metropolitan Museum of Art is in New York City, USA. Tokyo, Japan is over 6,000 miles away. So the answer is no. |
| OPENBOOKQA | INPUT (I): Our only star provides us with energy that is (a) temporary (b) inferior (c) expensive (d) reusable 
 OUTPUT (RO): The energy from the sun is renewable and reusable. So the answer is (d). |
| QUAREL | INPUT (I): Cutting bread with a bread knife creates a smooth cut, while cutting cake with a bread knife creates a rough cut. This means that the ____ has less resistance (A) bread (B) cake 
 OUTPUT (RO): When something has less resistance, it is easier to cut through. Thus, the bread has less resistance, making it easier to cut with the bread knife. So the answer is (A). |
| NUMERSENSE | INPUT (I): Fungi reproduce in <mask> ways. (A) no (B) zero (C) one (D) two (E) three (F) four (G) five (H) six (I) seven (J) eight (K) nine (L) ten 
 OUTPUT (RO): Fungi reproduce by sexual or asexual reproduction. So the answer is (D). |
| QASC | INPUT (I): Bees are necessary to (A) haploid plants (B) genetic diversity (C) spread flower seeds (D) prevent evolution (E) Reproduction (F) eat weeds (G) chew flowers (H) important habitats 
 OUTPUT (RO): Bees are necessary to spread flower seeds. Bees pollinate flowers, which helps the flowers reproduce and spread their seeds. So the answer is (C). |

BOOKQA (Mihaylov et al., 2018), NUMERSENSE Lin et al. (2020) and QASC Khot et al. (2020). These datasets were chosen over others which have existing human written rationales because all of them require certain level of implicit or logical reasoning in order to arrive at the answer. As we depict in the examples, we follow the **I-RO** format (Wiegreffe et al., 2021), wherein the input to the LM is the question, and the output is the joint generation of the rationale and the predicted answer.

In order to determine whether these generated rationales are of *good quality*, we focus on three properties that are necessary for any rationale to have, agnostic of the task it is meant for. First, we note that a rationale should be *plausible*. We define *plausibility* as the rationale making *sense* on its own – whether it be common, logical or factual sense depending on the dataset at hand. For example, if a rationale states 'Cows can fly', it is not plausible. Next, we identify that a rationale should be *diverse*, where the rationale is clean and not repetitive. Lastly, we note that a rationale should be *consistent* with the gold label for the input. *Consistency* is important to ensure that a rationale does not spew irrelevant information, and that it supports the gold answer. Furthermore, we focus on consistency with respect to the gold label, as misleading rationales are unhelpful as both LM justifications, and for human utility (Joshi et al., 2023). We formalise these properties as rewards in §4 and while these are necessary properties for any rationale, we also discuss other *good-to-have* properties in §5.

All of these properties are agnostic of the actual prediction made by the LM. Since our self-rationalization setup generates a rationale first, followed by its prediction, we aim to generate rationales with good quality, which should ideally improve the answer generated by the LM. Therefore, we focus on improving self-rationalization along these three properties, as well as on task accuracy. Along with the above rationale properties, we also consider *task correctness* as a necessary property of rationales, that they should try to improve over as a byproduct.

## 3   MARIO: OPTIMIZING FOR MULTIPLE REWARDS

To improve an LMs' rationalization across multiple properties, we leverage QUARK (Lu et al., 2022), a reinforcement learning-like framework effective on tasks such as unlearning toxicity in generations. We propose **M**ulti-rew**A**rd **R**at**IO**nalization (**MARIO**), an extension of QUARK to multiple rewards concurrently. We further propose two variants of MARIO: CLASSIC and ADDITIVE, that explore different ways of using QUARK for in a multi-reward setup. Figure 2 shows a running example of MARIO. Appendix B shows a second, more technical illustration of the same.

**QUARK**   QUARK (Lu et al., 2022) is a reinforcement learning-like algorithm that trains LMs using unique *control tokens* prepended to the generated text. The QUARK algorithm works iteratively:

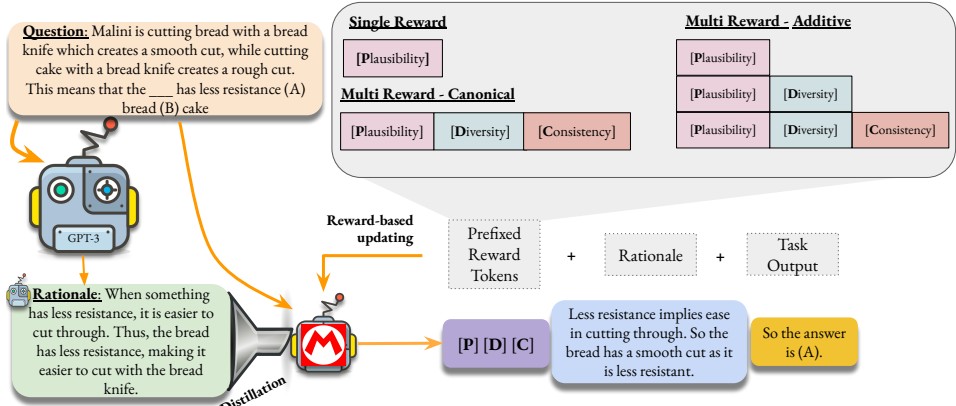

Figure 2: **MARIO pipeline.** MARIO uses rationales generated by a larger LM like GPT-3 as initial supervision, and uses rewards corresponding to three rationale properties: PLAUSIBILITY, DIVERSITY and CONSISTENCY, to improve self-rationalization of smaller LMs like T5-LARGE.

(1) sampling the a pre-trained LM to generate more training data, (2) scoring the generated data using a chosen reward metric, and (3) using instance-level scores to sort and bin the data into a fixed number of bins, each of which correspond to a unique control token. During training, the LM learns to associate each control token with the quality (as determined by the reward metric) of the data it is assigned to. During inference, in order to obtain the best quality generations, QUARK samples the LM using the control token corresponding to the highest reward measure. We provide a more detailed description of how QUARK works in Appendix B.

## 3.1 CLASSIC MARIO

Since QUARK is designed only for single reward optimizations, what if we are interested in improving the rationales along multiple rewards? In order to do this, we first propose a direct extension to the QUARK algorithm: instead of assigning each instance just one unique control token (which corresponds to one specific property), we now assign each instance *multiple* unique control tokens *at once*; now, each control token corresponds to a different property we want to train the LM on. We call this the **CLASSIC MARIO**. The *order* in which properties are represented in this method is a design decision that we can choose, and we expand on this further in Appendix C.

## 3.2 ADDITIVE MARIO

We now propose a step-by-step multi-reward QUARK: instead of training the LM on all the properties at the same time, we introduce them into training pipeline *additively*, in a predefined order of properties. For example, say that we have properties $P_1, P_2, P_3$, and that we want the LM to focus on $P_3$ first, before moving on to $P_1$ and then later $P_2$. In this method, we use multiple sets of control tokens as in CLASSIC MARIO, but, we introduce each set of tokens into the training pipeline successively. For example, we train the LM with only $P_3$ for the first $t$ steps, then we train the LM on $P_3$ and $P_1$ from the $t$-th step to the $2t$-th step, and then from the $2t$-th step onwards, we train the LM on $P_3$, $P_1$ and $P_2$. We call this method the **ADDITIVE MARIO**. Again, the order in which we add the control tokens of different rewards to the training pipeline, and whether each new control token is added to the left or right of the existing control tokens are decision choices, and we expand on it further in Appendix C.

## 4 EXPERIMENTS

### 4.1 DATASETS

As we mention in Section 2, We conduct experiments on 5 QA datasets: STRATEGYQA, QUAREL, OPENBOOKQA, NUMERSENSEand QASC; the task is to generate a rationale followed by the pre-

dicted answer. We report details of train, val and test splits in Appendix D. We do not require human-written gold rationales; we sample GPT-3 for silver rationales for supervision. We use chain-of-thought prompts from prior works on these datasets (refer Appendix I, E for the full prompts) and sample 5 rationale generations for each training set instance with a temperature of 0.9; for supervision we use only the instances where the answer predicted by GPT-3 is correct. We use these silver rationales in three places: (1) to train SFT, (2) we use SFT as the reference model for the KL divergence loss in MARIO's training process (Appendix B, E), (3) we also add these silver rationales to the overall data pool of MARIO (Appendix B provides an expanded explanation for the same).

## 4.2 RATIONALE PROPERTY REWARDS AND TASK CORRECTNESS

We formalize our chosen rationale properties (from §2) with their implementations that we use as rewards within MARIO. Let $Q$ be the input question, $\hat{R}$ represent the LM's generated rationale, and $O$ and $\hat{O}$ represent the gold answer and the LM's predicted answer respectively. The formulations of each reward are as follows:

- **PLAUSIBILITY** via VERA (Liu et al., 2023a): VERA is a trained commonsense statement verification T5-11B model, that provides a numerical score between $0$ and $1$, indicating the plausibility of declarative sentences.

$$\text{PLAUSIBILITY}(\hat{R}) = \text{VERA}(\hat{R}) \tag{1}$$

We use the VERA release from HuggingFace [3].

- **CONSISTENCY** via (Wiegreffe et al., 2021): Wiegreffe et al. (2021) provide a framework to evaluate the association between a rationale and a label with the help of two reference LMs that are trained to predict the answer with ($\mathbb{M}_{QR}$) and without ($\mathbb{M}_Q$) the rationale in the input. More formally,

$$\text{CONSISTENCY}(\hat{R}) = P_{\mathbb{M}_{QR}}(O|Q, \hat{R}) - P_{\mathbb{M}_Q}(O|Q) \tag{2}$$

CONSISTENCY is a numerical score between $-1$ and $1$ that indicates the faithfulness of the rationale towards the *gold answer*, like the implementation by Wiegreffe et al. (2021). Hyperparameters and training guidelines for the LMs involved in generating the CONSISTENCY score is in Appendix E.

- **DIVERSITY** via n-gram uniqueness in Li et al. (2022): Li et al. (2022) calculate diversity of generated text by determining the fraction of unique n-grams generated. DIVERSITY is a numerical score between $0$ and $1$ that indicates the lexical diversity of the rationale; this metric also serves the purpose of ensuring that the rationale does not contain repeated phrases or sentences.

$$\text{DIVERSITY}(\hat{R}) = \prod_{n=2}^{4} \frac{\text{unique n-grams}(\hat{R})}{\text{total n-grams}(\hat{R})} \tag{3}$$

- **TASK-CORRECTNESS**: As our last reward, we add task correctness of the answer that is generated following the rationale. This is a binary $0/1$ score, referring to the wrong and right predicted answer respectively.

$$\text{TASK-CORRECTNESS}(\hat{R}) = \mathbb{1}[\hat{O} = O] \tag{4}$$

We evaluate/report these metrics for all our baselines and experiments. To simplify comparisons, we also report an average normalized relative gain (NRG) (Chan et al., 2022) (Appendix E).

## 4.3 HUMAN PREFERENCE EVALUATION

We first present human preference studies comparing rationales generated by MARIO and the supervised fine-tuned baseline SFT for all five datasets. For each instance, we ask three distinct annotators from a pool of qualified annotators to compare the two rationales across three settings, for a given question and correct answer pair: PLAUSIBILITY and CONSISTENCY, which are defined in the

---

[3]`https://huggingface.co/liujch1998/vera`

Table 2: Demonstrative examples for the rationale properties

| QUESTION | Malini is cutting bread with a bread knife which creates a smooth cut, while cutting cake with a bread knife creates a rough cut. This means that the ___ has less resistance (A) bread (B) cake |
|---|---|
| PLAUSIBILITY | ⬇ Less resistance implies that the item would be *difficult* to cut through. Therefore, cake has less resistance. 
 ⬆ Less resistance implies ease in cutting through. So the bread has a smooth cut as it is less resistant. |
| CONSISTENCY | ⬇ Less resistance implies ease in cutting through. So the cake has a smooth cut as it is less resistant. 
 ⬆ Less resistance implies ease in cutting through. So the bread has a smooth cut as it is less resistant. |
| DIVERSITY | ⬇ Less resistance implies ease in ease in ease in cutting through. Ease in cutting through. Answer is bread. 
 ⬆ Less resistance implies ease in cutting through. So the bread has a smooth cut as it is less resistant. |

same manner as the rewards, and an overall PREFERENCE rating. PREFERENCE is meant to indicate that the annotators pick the rationale that they would find acceptable (Wiegreffe et al., 2022) for the given question. In Figure 3, we plot the % of instances where majority of annotators prefer only MARIO's rationales, only SFT's rationales, both or none. We note human annotators prefer MARIO's *only* rationales for 83.15%, 75.3%, 71.49%, 67.44% and 66.6% of instances respectively for STRATEGYQA, QUAREL OPENBOOKQA, NUMERSENSE and QASC. Human annotators also find MARIO's rationales to be considerably more plausible and consistent than SFT[4]. We use Amazon MTurk[5] for all our human studies, and Appendix J provides further details on the same.

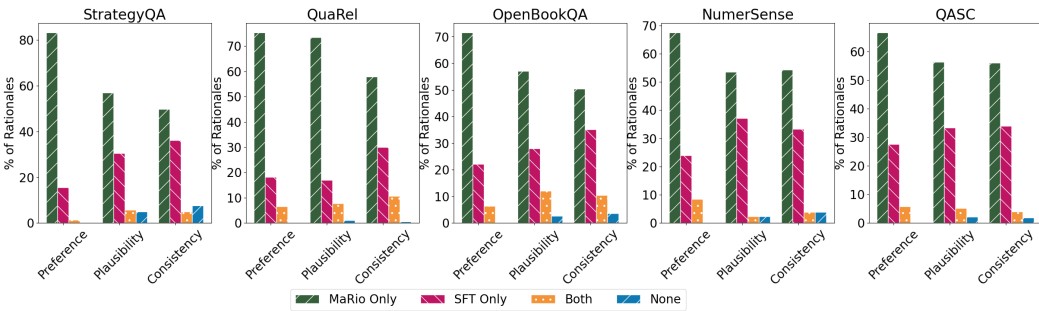

Figure 3: **Results of human studies comparing MARIO with SFT.** Here, we plot the % of instances in the test set wherein annotators prefer MARIO, SFT, both or none, with respect to PREFERENCE, PLAUSIBILITY and CONSISTENCY. We find that human annotators vastly prefer MARIO's rationales, and also find them to be much more plausible and consistent.

## 4.4 BASELINES VS. MARIO

All our baselines and MARIO are built on top of T5-LARGE LMs (0.7B). We present and compare our method with four strong baseline models:

1. Supervised Fine-tuned Self-Rationalizer (**SFT**): A fine-tuned T5-LARGE, which serves as the supervised learning baseline (we use the training data as described in §4.1), trained to generate rationales and answers.
2. Product of Rewards (**PRODUCT**): A multi-reward baseline where we consolidate the rewards into a single representative metric by taking their product and apply QUARK. Ag-

---

[4]We do not perform human studies for DIVERSITY and TASK ACCURACY since they are automatic/straightforward metrics

[5]https://www.mturk.com/

Table 3: **Baselines vs. MARIO Results.** For each dataset, the best averaged NRG (across TASK ACCURACY, PLAUSIBILITY, DIVERSITY and CONSISTENCY) is highlighted in **bold**, and each best individual metric is underlined. Cells marked with a * shows significant improvement for the corresponding MARIO configuration over SFT ($p < 0.05$).

| Method → | | Baselines | | | | MARIO | |
|---|---|---|---|---|---|---|---|
| Dataset ↓ | Metric | SFT | PRODUCT | FILT-ACC | FILT-ALL | CLASSIC | ADDITIVE |
| **STRATEGYQA** | Acc. | 57.64 | 62.01 | 61.57 | 61.35 | 60.26 | 65.07 |
| | Plau. | 0.33 | 0.35 | 0.34 | 0.36 | 0.38 | 0.39* |
| | Div. | 0.95 | 0.92 | 0.92 | 0.94 | 0.95 | 0.97* |
| | Cons. | -0.02 | 0.00 | 0.00 | 0.00 | 0.01 | 0.04* |
| | Avg. NRG | 58.66 | 59.75 | 59.39 | 60.34 | 60.94 | **63.27** |
| **QUAREL** | Acc. | 76.99 | 79.53 | 79.53 | 76.45 | 79.89 | 78.99 |
| | Plau. | 0.71 | 0.72 | 0.71 | 0.73 | 0.77* | 0.75 |
| | Div. | 0.95 | 0.95 | 0.95 | 0.95 | 0.97* | 0.97 |
| | Cons. | 0.18 | 0.21 | 0.20 | 0.17 | 0.19 | 0.20 |
| | Avg. NRG | 75.50 | 76.71 | 76.38 | 75.74 | **78.35** | 77.75 |
| **OPENBOOKQA** | Acc. | 63.65 | 61.65 | 65.86 | 56.63 | 66.06 | 65.55 |
| | Plau. | 0.53 | 0.52 | 0.55 | 0.47 | 0.55 | 0.55 |
| | Div. | 0.98 | 0.99 | 0.99 | 0.99 | 0.99* | 0.98 |
| | Cons. | 0.05 | 0.07 | 0.08 | 0.01 | 0.09* | 0.09 |
| | Avg. NRG | 66.79 | 66.54 | 68.47 | 63.28 | **68.64** | 68.29 |
| **NUMERSENSE** | Acc. | 46.23 | 50.75 | 51.76 | 46.73 | 55.28 | 54.27 |
| | Plau. | 0.60 | 0.60 | 0.61 | 0.58 | 0.63* | 0.63 |
| | Div. | 1.00 | 1.00 | 1.00 | 1.00 | 1.00 | 0.99 |
| | Cons. | 0.17 | 0.20 | 0.21 | 0.16 | 0.23* | 0.23 |
| | Avg. NRG | 66.18 | 67.69 | 68.32 | 65.68 | **69.95** | 69.44 |
| **QASC** | Acc. | 58.64 | 57.88 | 57.78 | 57.02 | 60.15 | 59.61 |
| | Plau. | 0.44 | 0.43 | 0.39 | 0.42 | 0.47* | 0.47 |
| | Div. | 0.96 | 0.95 | 0.96 | 0.96 | 0.99* | 0.99 |
| | Cons. | 0.19 | 0.17 | 0.17 | 0.17 | 0.19 | 0.19 |
| | Avg. NRG | 64.54 | 63.60 | 62.82 | 63.38 | **66.41** | 66.28 |

gregating several rewards into one is common in prior work and is often done through via product (Lu et al., 2023) or weighted average (Wu et al., 2023).

3. Filtering rationales that lead to correct answers (**FILT-ACC**): This is a variant of STAR (Zelikman et al., 2022). We iteratively train and sample new training data from a T5-LARGE, similar to QUARK, but instead of using any control tokens, we filter out the instances which have the wrong predicted label. We train this model with only cross-entropy loss.

4. Multi-reward variant of FILT-ACC (**FILT-ALL**): Again, we iteratively train and sample new training data from a T5-LARGE, and instead of using control tokens, we filter out the instances which have the wrong predicted label and instances that fall under a specified threshold value for PLAUSIBILITY, DIVERSITY and CONSISTENCY. The threshold value is tuned as a hyperparameter. We train this model with only cross-entropy loss.

Table 3 shows the comparisons between MARIO and the baselines. For all five datasets, we note that MARIO is the overall best setup as noted by both the individual metrics and the averaged NRG metric. ADDITIVE MARIO is found to be the best performing method for STRATEGYQA, and CLASSIC MARIO is found to be the best method for the other 4 datasets (hyperparameter configurations in Appendix E). It is important to note that not only does the rationale get better (as seen via the rationale metrics), but the task accuracy also shows a marked improvement over the baselines. We show some representative examples of rationales generated by training with MARIO, in comparison with those generated by SFT in Table 8. We also release the rationales generated by SFT and MARIO.[6]

---

[6]https://drive.google.com/drive/folders/1bWBxdiwce8US5y_
G6d9-Eki7ObllpR80?usp=sharing

### 4.5 REFERENCE LARGE LMS VS. MARIO

We now consider 3 strong reference LLMs that are used in practice for self-rationalization: GPT-3 (175B), FLAN-T5 (Chung et al., 2022) (sizes, L, XL, XXL) and LLAMA (Touvron et al., 2023) (sizes 7B, 65B); we compare MARIO with them in terms of both average NRG (Figure 4) and individual metric scores (Table 10). All these LMs apart from FLAN-T5-L are orders of magnitude larger than our T5-LARGE LM trained with MARIO; we include FLAN-T5-L in our comparison even though it's of the same size as MARIO because FLAN-T5-L is instruction-finetuned, and few-shot prompted to generate rationales, with the same set of demonstrations used by other large LMs (shown in Appendix I). Ideally, we want a small-sized LM (for efficiency) that achieves high performance, which corresponds to the top-left portion of the graph in Figure 4. Hence, to compare two LMs' performance, the LM which is relatively to the left *and* to the top is practically a better choice. We note that for QUAREL, MARIO results in an LM that is of a very small size (0.7B) but has a very high performance, almost equivalent to that of GPT-3. For NUMERSENSE, MARIO beats all models except for FLAN-T5-XXL and GPT-3, and for QASC, MARIO beats all models except for FLAN-T5-XXL, LLAMA-65B and GPT-3. For OPENBOOKQA , we see that MARIO beats LMs such as FLAN-T5-L, FLAN-T5-XL and LLAMA-7B. For STRATEGYQA we see that our LM beats FLAN-T5-L, while performing only a little worse than FLAN-T5-XL.

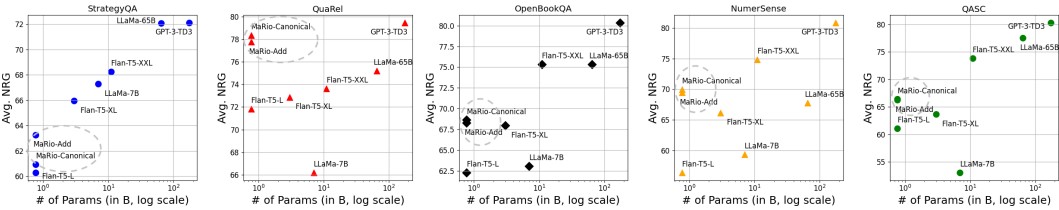

Figure 4: **Reference Large LMs vs. MARIO Results:** Here, we show the comparison of Avg. NRG values w.r.t the LM size (in the order of billion parameters) for all the datasets.

## 5 DISCUSSION

### 5.1 PROPERTIES AND METRICS

While the properties we explored in this work are *necessary* for high rationale quality, the question of what are the *complete* set of properties remains an open problem (Joshi et al., 2023; Wiegreffe et al., 2022; Golovneva et al., 2022). Some recent works on other necessary rationale properties are REV Chen et al. (2023a) (novelty of information, faithfulness towards the predicted label), ROSCOE Golovneva et al. (2022) / ReCEval Prasad et al. (2023) (score steps of reasoning), LAS Hase et al. (2020) (faithfulness towards predicted labels) etc. Further, there are also properties which do not have widespread implementations (to the best of our knowledge) such as factual-correctness, completeness of rationales (existing metrics require gold rationales which are not easily available, and which cannot score any alternate reasoning to the answer), etc. As future work, we hope to collect an extended set of properties and corresponding metrics, and improve them with MARIO.

### 5.2 MULTI-REWARD HACKING

As additional experimentation with alternate properties relevant to our chosen QA datasets, we worked on a set of experiments focusing on factual-correctness and lexical diversity; specifically for STRATEGYQA which requires historical or factual correctness of the rationale (this is different from common-sense or logical correctness measured by PLAUSIBILITY, as explained in VERA(Liu et al., 2023a)). We started with a fact verification metric LOREN (Chen et al., 2022) - while effective, we couldn't use this metric in practice since each score prediction required a Web API call, which is inefficient given MARIO's iterative data generation and scoring. We tried a weaker metric - querying the rationale with a larger LM, FLAN-T5-XXL and asking if the rationale was factually correct or not (probability of 'yes' under yes/no). We noticed that applying QUARK/MARIO with this metric led to some interesting reward hacking, as we show in the first two rows of Table 4.

Trying to improve on just factuality caused the LM to start generating repetitive text, as an easy way of generating factual statements. When we tried to counter this by training MARIO on factuality and lexical diversity together, the LM started generating incomplete rationales. We further noted that this kind of repetitive generation is observed even in larger LMs which on surface seem much better (as seen in rows 3 and 4 in Table 4). Therefore, we note that selecting strong rewards, as well as careful qualitative investigation is extremely important to prevent this kind of reward hacking – where an increase in individual numerical reward scores do not guarantee overall qualitative improvements.

Table 4: **Reward Hacking observed in training using QUARK:** We show examples of how if the rewards used are very weak or incompatible with each other, they can lead to strange reward hacking where the score numerically increase, but the rationales become worse qualitatively.

| Question and Rationale | Errors by other LMs |
|---|---|
| **Question:** Can a snow leopard swim? 
 **Rationale:** The snow leopard is a cat. The snow leopard is a cat. | T5-LARGE Repetition |
| **Question:** Would someone on Venus be unlikely to experience hypothermia? 
 **Rationale** Hypothermia is a condition in which the body is unable to produce enough heat. | T5-LARGE Incomplete |
| **Question:** Did Cleopatra have ethnicity closer to Egyptians than Greeks? 
 **Rationale:** Cleopatra was a member of the Roman dynasty of Egypt. 
 The Roman dynasty of Egypt was a branch of the Roman Empire. 
 The Roman Empire was a branch of the Roman Empire. | FLAN-T5-XL Repetition |
| **Question:** Is Fiat Chrysler gaining a new overall corporate identity? 
 **Rationale:** Fiat Chrysler is a brand of automobiles. Fiat Chrysler is a brand of cars. | FLAN-T5-XL Repetition |

## 5.3 IS ACCURACY ENOUGH OF AN INDICATOR FOR RATIONALE QUALITY?

As we discuss in §1, many contemporary works on self-rationalization ignore the independent quality of the generated rationale, and focus entirely on how the rationale can contribute to the task performance. In this discussion, we analyze the reverse: **if an LM is trained only with respect to task performance, what does this mean for the rationale?** We refer back to our main results, Table 3; we specifically look at the rows SFT, FILT-ACC and MARIO. We first see that both FILT-ACC and SFT both improve upon the TASK ACCURACY on all five datasets, as intended. We then see that for STRATEGYQA, QUAREL, NUMERSENSE and QASC, the average quality of the rationales generated by MARIO is decidedly better than the rationales generated by FILT-ACC, as seen by the values of the individual rationale quality metrics. For OPENBOOKQA, the analysis from just the metrics is inconclusive; hence, we perform human studies comparing FILT-ACC and MARIO, in the same manner as in §4.3. We find that human annotators prefer MARIO's rationales highly over that of FILT-ACC: for $69.65\%$ of the questions, majority of the annotators prefer MARIO's rationales (as opposed to $22.88\%$ of preference for FILT-ACC's rationales, and $7.46\%$ preference for *both*). We further performed human studies for PLAUSIBILITY and CONSISTENCY, and again, MARIO's rationales were found to be distinctly better (PLAUSIBILITY: $49.5\%$ preference for MARIO, $32.58\%$ for SFT, $13.43\%$ both, $0.99\%$ neither; CONSISTENCY: $48\%$ preference for MARIO, $37.31\%$ for SFT, $9.45\%$ both, $2.48\%$ neither). In conclusion, we find that optimizing for task performance does not naturally improve rationale performance, which further motivates the introduction of MARIO.

## 6 CONCLUSION AND FUTURE WORK

Existing self-rationalization LMs use rationales as a means for improving downstream task accuracy, with the help of large-scale LMs. In this work, we propose MARIO, an algorithm that performs multi-reward optimization of small self-rationalizing LMs to jointly improve the quality of their rationales as well as their task accuracy. We present strong experimental results on a small LM, T5-LARGE, over competitive baselines, on datasets STRATEGYQA, QUAREL OPENBOOKQA, NUMERSENSE and QASC. In addition to a strong improvement in task accuracy, we see that rationales produced by training an LM with our method are strongly preferred by human annotators. Lastly, we discuss intricacies of reward-conditioned rationale generation for small LMs, issues faced with selecting appropriate rewards, as well as shortcuts taken by QUARK to improve reward scores that do not translate well to qualitative improvement. As future work, we hope to extend our algorithm to improving rationales along more dimensions like completeness, factuality as well as human utility.

## ACKNOWLEDGEMENTS

This research is supported in part by the Office of the Director of National Intelligence (ODNI), Intelligence Advanced Research Projects Activity (IARPA), via the HIATUS Program contract #2022-22072200006 and the USC + Amazon Center on Secure & Trusted Machine Learning. The views and conclusions contained herein are those of the authors and should not be interpreted as necessarily representing the official policies, either expressed or implied, of ODNI, IARPA, or the U.S. Government. The U.S. Government is authorized to reproduce and distribute reprints for governmental purposes notwithstanding any copyright annotation therein. We would like to thank all of our collaborators at the USC NLP Group and USC INK Research Lab for their constructive feedback on this work. We also thank the reviewers for their valuable and constructive suggestions.

## ETHICAL CONSIDERATIONS

Like any natural language generation system/algorithm, MARIO can unintentionally lead to toxic and harmful text; it is up to the user of the algorithm to use it responsibly, with non-harmful reward metrics, to prevent the generation of biased and malicious outputs. As noted in McGuffie & Newhouse (2020), this is a deliberate misuse of text generation models, and we strongly denounce such practices.

**Data.** All the datasets that we use in our work are released publicly for usage and have been duly attributed to their original authors.

**Crowdsourcing.** All our crowdworkers are from countries where English is the primary language. For all our human studies, the task is setup in a manner that ensure that the annotators receive compensation that is above minimum wage ($20/hour). Since we conduct extensive qualification tasks before annotations, crowdworkers that participate in the qualification are compensated more than the task, given the time taken to read and understand task instructions and examples. Furthermore, we ensure that we correspond with crowdworkers over email to address their queries. Crowdworkers have also been given bonuses for flagging errors in the task, or consistently providing good-quality annotations.

## REPRODUCIBILITY

For all our experimental results and models, we report (1) the complete hyperparameter setting and any bounds explored (Appendix E) as well as the sizes and versions/pretrained-model links of all models used, (2) the time taken per experiment, and infrastructure used, (3) the mathematical equations (§4.2, Appendix B) for all algorithms and metrics used, (4) descriptions of datasets, and demonstrations used to sample rationales from GPT-3. All our codes and datasets are publicly released at `https://github.com/INK-USC/RationaleMultiRewardDistillation`.

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

# A   RELATED WORK

**Self-rationalization and rationale-based distillation.** Model decisions can be explained in two ways - by extracting rationales from the input text, or generating free-text rationales that may not be grounded in the input. An extractive rationale explains a model's output on a given task instance by scoring input tokens' influence on the model's output (Denil et al., 2014; Sundararajan et al., 2017; Li et al., 2016; Jin et al., 2019; Lundberg & Lee, 2017; Chan et al., 2022). This token scoring can be done via input gradients (Sundararajan et al., 2017; Lundberg & Lee, 2017; Denil et al., 2014; Li et al., 2015), input perturbation (Li et al., 2016; Poerner et al., 2018; Kádár et al., 2017), attention weights (Pruthi et al., 2020; Stacey et al., 2022; Wiegreffe & Pinter, 2019), or learned rationale extraction models (Lei et al., 2016; Chan et al., 2022; Jain et al., 2020; Situ et al., 2021; Liu et al., 2023b). For the purpose of this work, we mainly focus on free-text rationales. There are two primary methods adopted by prior works for generating free-text rationales. The first set of approaches use gold human-written rationales to train a rationale generation model (Camburu et al., 2018; Narang et al., 2020; Wiegreffe et al., 2021). The second set of approaches prompt large LMs with the help of curated templates with or without demonstrations containing examples of rationale generation for the task at hand (Wei et al., 2022; Kojima et al., 2023; Li et al., 2023c; Jung et al., 2022; Lightman et al., 2023). Some approaches also leverage few-shot training approaches with a handful of gold rationales (Marasovic et al., 2022; Chen et al., 2023b). Recent approaches also leverage rationales generated by large LMs to distill small LMs to be better at the task or better rationalizers. (Pruthi et al., 2022; Li et al., 2023b; Chan et al., 2023; Wang et al., 2023b; Saha et al., 2023; Hsieh et al., 2023)

**Evaluating free-text rationales.** Existing works have conducted human and automatic evaluation of free-text rationales based on their association with predicted labels (Wiegreffe et al., 2021), acceptability (Wiegreffe et al., 2022), informativeness (Chen et al., 2023a), benefits and human utility (Sun et al., 2022; Joshi et al., 2023), simulatability (Rajani et al., 2019; Hase et al., 2020) and faithfulness (Atanasova et al., 2023; Wang et al., 2023a) to name a few. Some recent works have also provided frameworks to evaluate logical correctness of reasoning chains, that are similar to free-text rationales (Golovneva et al., 2022; Prasad et al., 2023).

**Reward-conditioned text generation.** Reinforcement learning has proven to be a reliable means to optimize language models towards a specific objective. One such example, proximal policy optimization (PPO) (Schulman et al., 2017), has been commonly used for a variety of tasks, spanning detoxification (Wu et al., 2023; Lu et al., 2022), RLHF (Dubois et al., 2023; Bai et al., 2022), improving commonsense reasoning capabilities (Liu et al., 2022), and more. Adjacent to PPO, there are several other lighter-weight algorithms which condition the policy language model *directly* on the reward without the need for a value function (Lu et al., 2022; Gulcehre et al., 2023; Dong et al., 2023; Lu et al., 2023; Zelikman et al., 2022). These methods rely on iterative, off-policy explorations at fixed intervals to continuously aggregate new trajectories to learn from. Another line of work improves the reward directly through iterative refinement on a frozen policy model (Madaan et al., 2023). There are several algorithms and methods today to update text generation models with rewards. Lu et al. (2022) that unlearns toxicity by specifically fine-tuning the model on what *not* to do, Lu et al. (2023) which tailors the generation of extremely large LMs like GPT-3 using trained policy adaptor models. Zelikman et al. (2022) that leverages a small number of demonstrations to iteratively generate new data to train the model (new data such that the task prediction is correct). Other recent work on controllable text generation revolves around creative text generation with single and multiple rewards (Yang & Klein, 2021; Keskar et al., 2019; Zhang et al., 2023; Qian et al., 2022; Yang et al., 2022)

# B   QUARK AND MARIO

Here, we describe QUARK and MARIO in more technical detail (refer the top and bottom pipelines in Figure 5 respectively).

QUARK begins training with a pretrained trained language model $P_0(t|x)$; QUARK also requires a *reference* language model $P_{ref}(t|x)$ (which can be the same as $P_0$, or different), and a reward function $Rew(t, x) \rightarrow \mathbb{R}$. Note that $x = [x_0, x_1, \ldots, x_{m-1}]$ stands for the input text sequence, and $t = [t_0, t_1, \ldots, t_{n-1}]$ stands for the output sequence generation. Lastly, QUARK works with a

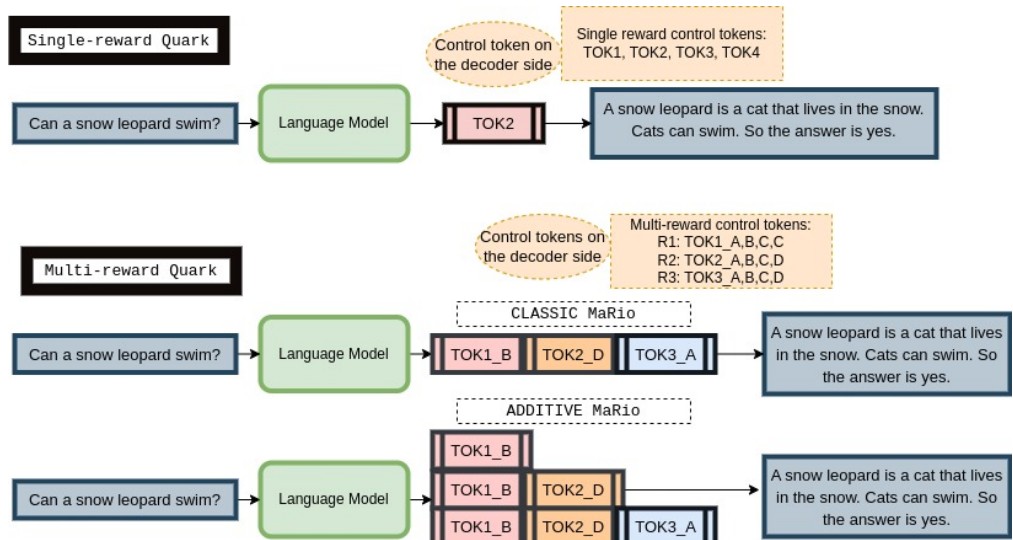

Figure 5: Optimizing properties with QUARK (top) and MARIO (bottom)

data pool $D$ which is constantly updated and added to over the course of training (as we describe below); further, $D$ can be initialized with gold-standard or silver-standard data, $D = D_{gold/silver} = (x, t_{gold/silver}, r)$.

As we explain in Section 3, QUARK operates in an iterative fashion:

1. sampling $P_0$ to generate more training data: $D_{new} = (x, t_{new})$

2. scoring the generated data using $Rew(x, t)$: $D'_{new} = (x, t_{new}, r)$

3. using these instance-level scores to sort and bin the data into a fixed number of bins $[b_1, b_2, \ldots, b_5]$ each of which correspond to a unique control token: $D''_{new} = (x, t_{new}, r, b)$,

4. Adding the now control-token attached data to the (growing) training data pool: $D = D \cup D''_{new}$

During training, the model starts to associate each control token with its corresponding quality of data (as given by $Rew(x, t)$), and to obtain the best quality generations during inference, QUARK samples the trained $P_0$ using the control token corresponding to the highest reward measure. QUARK is trained using the following training objectives:

- **Reward-based learning** using implicit reward signals based on control tokens (which are obtained by sorting the reward $Rew(x, t)$ scores), as described above,

- **Language model objective** using supervised/cross-entropy loss with respect to the target generation (as explained above, QUARKsamples training data in an online manner from $P_0$; however, if gold or silver offline training data is available, that can also be injected into the training pipeline by scoring with $Rew(x, t)$)

- **Stable text generation** using the KL divergence penalty of $P_0$'s generation with respect to $P_{ref}$, and.

- **Entropy regularization** of the generated text as in Meister et al. (2020)

The objective function for QUARK is:

$$\max_{\theta} \mathbb{E}_{k \sim \mathcal{U}(1, K)} \mathbb{E}_{(x,y) \sim \mathcal{D}^k} \left[ \log p_{\theta}(y \mid x, r_k) - \beta \sum_{t=1}^{T} \mathrm{KL}\left(p_0(\cdot \mid y_{<t}, x) \,\|\, p_{\theta}(\cdot \mid y_{<t}, x, r_k)\right) \right] \quad (5)$$

Here, the first term stands for the supervised cross-entropy loss, and the second term stands for the KL divergence loss. Entropy regularization can also be added if/when needed. Note that $x$ is the input text, $y$ is the generated output sequence and $r_k, k \in \{1, \ldots, K\}$ stands for the reward/control token.

We extend QUARK to MARIO by using multiple sets of control tokens, each corresponding to a distinct reward/property, i.e., $Rew_1(x, t), Rew_2(x, t), \ldots, Rew_k(x, t)$; the CLASSIC and ADDITIVE methods use these control tokens either together, or in a step-by-step fashion as we explain in §3.1, 3.2. Further, we want to note that step-3 of the algorithm (wherein we use instance-level scores to sort and bin the data) is done in MARIO separately for each reward; each reward/property goes through an individual scoring + binning process and gets a distinct control token. Subsequently, each reward/property also has its own set of control tokens (as depicted in Figure 5). The rest of the training follows the same iterative process and training objectives as QUARK. The objective function for MARIO is:

$$\max_\theta \mathbb{E}_{j \sim \mathcal{U}(1, J), k \sim \mathcal{U}(1, K)} \mathbb{E}_{(x,y) \sim \mathcal{D}^k}$$

$$\left[ \log p_\theta \left( y \mid x, [.., r_{jk}, ..] \right) - \beta \sum_{t=1}^{T} \mathrm{KL} \left( p_0 \left( \cdot \mid y_{<t}, x \right) \| p_\theta \left( \cdot \mid y_{<t}, x, [.., r_{jk}, ..] \right) \right) \right] \quad (6)$$

Here again, the first term stands for the supervised cross-entropy loss, and the second term stands for the KL divergence loss; entropy regularization can be added if/when needed. $x$ is the input text, $y$ is the generated output sequence and $r_{jk}, j \in \{1, \ldots, J\}, k \in \{1, \ldots, K\}$ stands for the reward/control token corresponding to the $j$-th property and the $k$-th reward bin.

## C    ORDER OF TOKENS

As we explain in the above two sections, the order of the control tokens corresponding to each reward we use in training our self-rationalizing LM is a design choice. Say for example, we have three properties, along with control tokens corresponding to the task accuracy (as we do in this paper, refer §4.2): this means that there are potentially 24 orders of these properties that we can use in CLASSIC MARIO, and 48 possible variations that we can use for ADDITIVE MARIO (24 orders x 2 directions in which we can introduce the property to the training – left or right of the existing control tokens, assuming we keep the direction of addition consistent throughout training). It is impractical and inefficient to experiment with all these possible orders to pick the best possible one. Hence, we propose a simple way of picking the order, based on the relative strengths of a (supervised-trained) self-rationalizing LM in each of these properties.

For example, say we have four reward metrics $R_1, R_2, R_3, R_4$, and we determine through a predefined method which property the LM is relatively stronger in (for example, say the LM is good at generating lexically diverse statements, but is only moderately good at grammar, is broadly bad at generating plausible statements, and even worse at producing concise rationales). For example, we determine the relative strength of rewards based on how good the supervised finetuned baseline SFT *is* on a particular metric on the validation set, as opposed to the maximum and minimum value of the metric itself.

$$\mathrm{strength}(R_i) = \frac{\max(R_i) - r_i}{\max(R_i) - \min(R_i)} \quad (7)$$

Here $R_i$ refers to the reward, $r_i$ refers to the value the SFT has on the property $R_i$ on the validation set, and max/min$(R_i)$ refer to the maximum and minimum value taken by the reward metric $R_i$.

For example, let the relative order of the four reward metrics using the above approach is $R_2 < R_1 < R_4 < R_3$. Hence, we experiment with training the LM with the order $R_2, R_1, R_4, R_3$ if we want to allow the weaker rewards to improve on their own, before the stronger rewards are introduced into the mix. Additionally, we can also use the opposite order $R_3, R_4, R_1, R_2$, so that the LM can quickly optimize on the stronger rewards and then try to be better with the weaker rewards.

# D    DATASET SPLITS

- For STRATEGYQA, since labels are not available for evaluation sets, we split the train set into training, validation and test sets (taken from Joshi et al. (2023)), and report scores on this test set.

- For OPENBOOKQA and QUAREL, we use the provided training dataset, tuned on the validation set and report final performances on the test set[7].

- For NUMERSENSE, we use the train, validation and test sets as in the official GitHub[8] release.

- For QASC, we split the original train set into train and validation (900 questions chosen randomly for validation), and use the original validation set as the test set[9].

All datasets have multi-choice questions (yes/no for STRATEGYQA, a/b for QUAREL, a/b/c/d for OPENBOOKQA, a/b/-/l for NUMERSENSE, a/b/-/h for QASC), and the task is to generate a rationale followed by the predicted answer.

# E    HYPERPARAMETERS AND EVALUATION

We use T5-LARGE ($0.7B$ parameters) for SFT and all our MARIO experiments, and we use T5-BASE for our CONSISTENCY models (as used in the original work Wiegreffe et al. (2021)) - we always start training with the pretrained model from HuggingFace[10]. Tables 5, 6 and 7 show the hyperparameters used to train SFT, CONSISTENCY and MARIO respectively. Note that for our MARIO training, we use SFT as the reference model ($P_{ref}(t|x)$ from Appendix B) for the KL divergence penalty. We also use the silver rationales sampled from GPT-3 as our initial data pool $D$ (from Appendix B). Further, during inference, we always use greedy decoding. We run all our experiments on NVIDIA Quadro RTX 8000 GPUs. For training SFT and CONSISTENCY models, we use 1 GPU per experiment; for training MARIO, we use 2 GPUs per experiment - the first GPU to hold $P_0, P_{ref}$ (notation from Appendix B), and the second GPU to hold the PLAUSIBILITY and CONSISTENCY reward models.

Furthermore, we aggregate metrics using Normalized Relative Gain as mentioned in Chan et al. (2022). NRG of a metric value $z_i$ (corresponding to the general property $Z$) is formally defined as:

$$\text{NRG}(z_i) = \frac{z_i - \min(Z)}{\max(Z) - \min(Z)} \tag{8}$$

The average NRG of a set of metrics (such as with the four metrics in this work) is a simple mathematical average of their individual NRG's.

Table 5: SFT training details

| Hyperparameter | Value |
|---|---|
| Optimizer | Adam |
| Adam epsilon | $1e$-8 |
| Adam initial learning-rate | $3e$-5 |
| Learning-rate scheduler | linear with warmup |
| Warmup steps | 1000 |
| Gradient clipping | 0.5 |
| Train batch-size | 4 / 8 |
| Training time | ~ 4 hours on 1 GPU |

Further, for our statistical significance tests, are done using one-tailed independent t-tests (using `scipy.stats.ttest_ind`).

---

[7]`https://huggingface.co/datasets/openbookqa,https://huggingface.co/datasets/QuaRel`

[8]`https://github.com/INK-USC/NumerSense/tree/main/data`

[9]`https://huggingface.co/datasets/qasc`

[10]`https://huggingface.co/t5-large,https://huggingface.co/t5-base`

Table 6: Training details for the $\mathbb{M}_{QR}$ and $\mathbb{M}_Q$ models used for CONSISTENCY

| Hyperparameter | Value |
|---|---|
| Optimizer | Adam |
| Adam epsilon | 1*e*-8 |
| Adam initial learning-rate | 3*e*-5 |
| Learning-rate scheduler | linear with warmup |
| Warmup steps | 1000 |
| Gradient clipping | 0.5 |
| Train batch-size | 4 / 32 |
| Training time | ~ 4 hours on 1 GPU |

Table 7: QUARK and MARIO training details

| Hyperparameter | Value |
|---|---|
| Optimizer | Adam |
| Adam epsilon | 1*e*-8 |
| Adam initial learning-rate | 3*e*-5 |
| Learning-rate scheduler | linear with warmup |
| Warmup steps | 1000 |
| Gradient clipping | 1.0 |
| Gradient accumulation | 2 steps |
| KL-divergence coef. | 0.05 / 0.1 |
| Entropy regularization coef. | 0.05 / 0.0 |
| Sampling rate | 1 (QUAREL, NUMERSENSE, QASC) or 2 (STRATEGYQA, OPENBOOKQA) samples for every train sample |
| Frequency of exploration | every 300 (STRATEGYQA, QUAREL) / 4000 (OPENBOOKQA, NUMERSENSE, QASC) steps |
| Sampling strategy | Top-p (0.7) sampling |
| Temperature for sampling | 1.0 |
| Number of distinct reward-bins | 5 for rationale metrics, 2 for TASK ACCURACY |
| Train batch-size | 4 |
| Training time | ~ 1 day on 2 GPUs |
| Order of rewards | STRATEGYQA: strongest to weakest, add to right QUAREL: strongest to weakest, OPENBOOKQA: weakest to strongest, NUMERSENSE: weakest to strongest, QASC: strongest to weakest |

# F    REPRESENTATIVE EXAMPLES OF RATIONALES

Table 8 shows some examples of rationales generated by MARIO.

# G    SINGLE REWARD EXPERIMENTS

For completeness of analysis, we present single-reward QUARK experiments, where we focus on improving just one property. Table 9 shows results on the same. We first note that in most of the cases, MARIO achieves an equivalent or better improvement as compared to single-reward QUARK. Further, we note that even if individually some properties are better when trained under single reward QUARK as compared to MARIO, MARIO is the only experiment where *all* the properties improve as compared to the SFT baseline. We also see that sometimes, single reward QUARK leads to improvement in other metrics as well; this could be because the metrics are positively correlated for that dataset. However, since we want to improve all metrics comprehensively, MARIO is a deterministic

Table 8: **Examples of Rationales:** Here, we show rationales generated by MARIO and SFT for some examples for the three datasets. We observe that sometimes, SFT generates longer rationales which are not consistent, or logically correct, which is fixed by MARIO.

| Dataset | Examples |
|---|---|
| STRATEGYQA | **Question**: While on a liquid diet, are there some types of soup you cannot eat? Yes or No?
**MARIO Rationale**: While on a liquid diet, there are some types of soup you cannot eat, such as cream of mushroom soup, which is a type of savory soup.
**SFT Rationale**: While on a liquid diet, there are certain types of soups that are not suitable for a liquid diet such as a liquid diet. |
| | **Question**: Is a beard is moss that grows on a human? Yes or No?
**MARIO Rationale**: A beard is a type of hair that grows on a human.
**SFT Rationale**: A beard is a type of facial hair that grows from a root and is typically wavy to dark in color. It is not moss, but a plant that provides shelter to the moss and helps it grow. |
| QUAREL | **Question**: A bus and a bicycle are both going the same direction. The bus is going 40mph faster than the bicycle. Will the bicycle in one hour cover (A) more distance (B) less distance
**MARIO Rationale**: The bus is going faster than the bicycle. Thus, the bicycle will cover less distance in one hour.
**SFT Rationale**: When something is going faster, it covers more distance. Thus, the bicycle will cover more distance in one hour because it is going faster than the bus. |
| | **Question**: Jim and Joe are riding bikes. Jim stops after fifteen minutes and takes a break. Joe keeps on riding. Who went farther? (A) Jim (B) Joe
**MARIO Rationale**: When someone is riding for a longer period of time, they will go farther. Jim stopped after fifteen minutes and took a break, while Joe kept on riding. Thus, Joe went farther.
**SFT Rationale**: When someone is moving, they will go farther than someone who is not. Thus, Joe went farther because he kept on riding. |
| OPENBOOKQA | **Question**: In order for crops to grow food safely, pesticides are used on them. When it floods, this causes t he what to be poisonous? (a) air (b) Corn (c) Runoff (d) farmers
**MARIO Rationale**: When it floods, runoff from the crops is sprayed with pesticides, making them poisonous.
**SFT Rationale**: When it floods, pesticides are sprayed onto crops, which can make them poisonous. |
| | **Question**: Plant growth may cause (a) an uptick in the number of leaves (b) a surge in leaf disease (c) a gradual decrease in leaves (d) a rapid decline of the leaves
**MARIO Rationale**: Plant growth may cause an uptick in the number of leaves.
**SFT Rationale**: Plant growth is a process of adding new leaves to the plant. This process can cause a gradual decrease in the number of leaves. |
| NUMERSENSE | **Question**: People have no more than <mask> legs. (A) no (B) zero (C) one (D) two (E) three (F) four (G) five (H) six (I) seven (J) eight (K) nine (L) ten
**MARIO Rationale**: People have two legs.
**SFT Rationale**: People have four legs. |
| | **Question**: One in <mask> also means a quarter. (A) no (B) zero (C) one (D) two (E) three (F) four (G) five (H) six (I) seven (J) eight (K) nine (L) ten
**MARIO Rationale**: One in four is a quarter.
**SFT Rationale**: One in a quarter. |
| QASC | **Question**: What measures a meter? (A) meter stick (B) football field (C) tetraceratops (D) visibility (E) weather (F) U.S. customary units (G) horses (H) lamphreys
**MARIO Rationale**: The metric system is based on the idea that a meter is equal to 100 centimeters.
**SFT Rationale**: A meter is a unit of length. The most common unit of measurement used in the United States is the meter. |
| | **Question**: What can be used to navigate? (A) music (B) backbones (C) astrology (D) cilia (E) magic (F) sponges (G) astronomy (H) lamphreys
**MARIO Rationale**: Cilia are tiny hairs on the surface of the skin that can be used to navigate.
**SFT Rationale**: Navigating requires using a variety of tools and techniques. Some of these tools include compass, compass, compass, compass, compass, compass, compass, compass, compass, compass, compass, compass, compass, compass, compass, compass, compass, compass, compass, compass, compass, compass, and com |

way to achieve the same. (Note: We don't run the experiment on NUMERSENSE DIVERSITY, since SFT already achieves the best possible value of 1.0).

## H EXTENDED COMPARISON WITH FEW-SHOT LLMS

In Table 10, we present the detailed performance metrics of different reference LMs as opposed to MARIO. For QUAREL, MARIO beats all reference LLMs except for GPT-3 on all four metrics. For NUMERSENSE, MARIO beats all reference LLMs except for FLAN-T5-XXL and GPT-3 on all four metrics. The results are more varied with STRATEGYQA, OPENBOOKQA and QASC; MARIO is better than the reference LLMs (apart from GPT-3) in the case of DIVERSITY for all three datasets, and in cases of varying comparisons with the reference LLMs (for example, MARIO is better at CONSISTENCY than FLAN-T5-L and LLAMA-7B for OPENBOOKQA). However, overall, we note that our model still needs to go further with respect to PLAUSIBILITY and TASK ACCURACY. We note that our method MARIO has done a significant job in bridging the gap between LMs such as the ones discussed in this section, and much smaller LMs such as T5-LARGE. We also note for TASK ACCURACY, CONSISTENCY and DIVERSITY, MARIO beats FLAN-T5-L, a model of equal size which has been trained with instruction fine-tuning for all 5 datasets (except for QASC and CONSISTENCY); and for all datasets except for STRATEGYQA, MARIO also beats PLAUSIBILITY of FLAN-T5-L.

Table 9: QUARK experiments on improving single rewards. For each dataset, the best averaged NRG (across TASK ACCURACY, PLAUSIBILITY, DIVERSITY and CONSISTENCY) is highlighted in **bold**, and each best individual metric is underlined.

| Method → | | Baselines | | Single Reward QUARK | | | | MARIO | |
|---|---|---|---|---|---|---|---|---|---|
| Dataset ↓ | Metric | SFT | PRODUCT | Acc. | Plau. | Div. | Cons. | CLASSIC | ADDITIVE |
| **STRATEGYQA** | Acc. | 57.64 | 62.01 | 61.57 | 61.35 | 59.17 | 59.17 | 60.26 | 65.07 |
| | Plau. | 0.33 | 0.35 | 0.36 | 0.36 | 0.36 | 0.36 | 0.38 | 0.39 |
| | Div. | 0.95 | 0.92 | 0.92 | 0.93 | 0.96 | 0.95 | 0.95 | 0.97 |
| | Cons. | -0.02 | 0.00 | -0.01 | 0.01 | -0.04 | 0.01 | 0.01 | 0.04 |
| | Avg. NRG | 58.66 | 59.75 | 59.77 | 60.21 | 59.79 | 60.17 | 60.94 | **63.27** |
| **QUAREL** | Acc. | 76.99 | 79.53 | 81.88 | 80.62 | 78.99 | 80.62 | 79.89 | 78.99 |
| | Plau. | 0.71 | 0.72 | 0.74 | 0.81 | 0.71 | 0.73 | 0.77 | 0.75 |
| | Div. | 0.95 | 0.95 | 0.95 | 0.93 | 0.97 | 0.95 | 0.97 | 0.97 |
| | Cons. | 0.18 | 0.21 | 0.23 | 0.20 | 0.20 | 0.22 | 0.19 | 0.20 |
| | Avg. NRG | 75.50 | 76.71 | 78.1 | **78.66** | 77.0 | 77.41 | 78.35 | 77.75 |
| **OPENBOOKQA** | Acc. | 63.65 | 61.65 | 64.46 | 61.65 | 64.66 | 66.27 | 66.06 | 65.55 |
| | Plau. | 0.53 | 0.52 | 0.54 | 0.53 | 0.51 | 0.54 | 0.55 | 0.55 |
| | Div. | 0.98 | 0.99 | 0.99 | 0.99 | 0.99 | 0.99 | 0.99 | 0.98 |
| | Cons. | 0.05 | 0.07 | 0.09 | 0.07 | 0.07 | 0.11 | 0.09 | 0.09 |
| | Avg. NRG | 66.79 | 66.54 | 67.99 | 66.79 | 67.04 | **68.69** | 68.64 | 68.29 |
| **NUMERSENSE** | Acc. | 46.23 | 50.75 | 51.76 | 50.75 | - | 54.27 | 55.28 | 54.27 |
| | Plau. | 0.60 | 0.60 | 0.63 | 0.63 | - | 0.61 | 0.63 | 0.63 |
| | Div. | 1.00 | 1.00 | 0.99 | 1.00 | - | 1.00 | 1.00 | 0.99 |
| | Cons. | 0.17 | 0.20 | 0.21 | 0.21 | - | 0.22 | 0.23 | 0.23 |
| | Avg. NRG | 66.18 | 67.69 | 68.57 | 68.56 | - | 69.07 | **69.95** | 69.44 |
| **QASC** | Acc. | 58.64 | 57.88 | 58.21 | 57.88 | 58.1 | 58.75 | 60.15 | 59.61 |
| | Plau. | 0.44 | 0.43 | 0.42 | 0.45 | 0.40 | 0.41 | 0.47 | 0.47 |
| | Div. | 0.96 | 0.95 | 0.96 | 0.97 | 0.98 | 0.96 | 0.99 | 0.99 |
| | Cons. | 0.19 | 0.17 | 0.17 | 0.17 | 0.17 | 0.20 | 0.19 | 0.19 |
| | Avg. NRG | 64.54 | 63.60 | 63.68 | 64.6 | 63.65 | 63.94 | **66.41** | 66.28 |

## I  FEW-SHOT DEMONSTRATIONS

We include the full few-shot demonstrations used to prompt different models for three datasets in Tables 11-13. For clarity, the rationalizations are highlighted.

## J  CROWDSOURCING FOR HUMAN EVALUATIONS

In this section, we describe the MTurk experiment setup. Each MTurk annotator is paid above minimum wage. Since the dataset we used is carefully annotated by human, we can assure there is no toxic content and our experiment setup was submitted to IRB for ethical review. We limited our Turkers to English speaking nations - United States, Canada, Australia, New Zealand and United Kingdom.

To ensure the quality of evaluation, we conduct a round of qualification tasks which include a small set of evaluations. Turkers need to finish the qualification task first and get results of it, then we will show them the whole task.

### J.0.1  WORKER SELECTION AND QUALITY CONTROL

Here, we describe details about how workers are selected and how annotations are ensured to be clean. First, we employ multiple rounds of trials before deploying the actual task so as to get feedback from annotators whether they understand the task correctly. This includes in-house tests, tested via Amazon Turk Sandbox [11] and small batches tested on Turk. Second, we create a set of medium to hard qualification tasks for verifying preference, plausibility and consistency annotations

---

[11] https://requester.mturk.com/developer/sandbox

Table 10: We compare MARIO with strong few-shot reference LMs: FLAN-T5, LLAMA and GPT-3. Apart from FLAN-T5-L (which we have included to show a model of equivalent size that has been instruction finetuned), all these models are much bigger than our T5-LARGE trained with MARIO.

| Method → | | FLAN-T5 | | | LLAMA | | GPT-3 | MARIO (0.7B) | |
|---|---|---|---|---|---|---|---|---|---|
| Dataset ↓ | Metric | L | XL | XXL | 7B | 65B | T-D-003 | CLASSIC | ADDITIVE |
| **STRATEGYQA** | Acc. | 54.59 | 71.83 | 70.52 | 59.17 | 72.27 | 69.0 | 60.26 | 65.07 |
| | Plau. | 0.49 | 0.59 | 0.64 | 0.72 | 0.70 | 0.70 | 0.38 | 0.39 |
| | Div. | 0.88 | 0.82 | 0.86 | 0.88 | 0.93 | 0.95 | 0.95 | 0.97 |
| | Cons. | -0.01 | 0.02 | 0.05 | 0.00 | 0.06 | 0.09 | 0.01 | 0.04 |
| | Avg. NRG | 60.27 | 65.96 | 68.26 | 67.29 | 72.07 | 72.13 | 60.94 | 63.27 |
| **QUAREL** | Acc. | 77.36 | 76.99 | 77.54 | 56.70 | 76.27 | 83.33 | 79.89 | 78.99 |
| | Plau. | 0.60 | 0.68 | 0.70 | 0.64 | 0.70 | 0.78 | 0.77 | 0.75 |
| | Div. | 0.93 | 0.90 | 0.92 | 0.94 | 0.96 | 0.95 | 0.97 | 0.97 |
| | Cons. | 0.14 | 0.13 | 0.10 | 0.00 | 0.17 | 0.23 | 0.19 | 0.20 |
| | Avg. NRG | 71.84 | 72.87 | 73.64 | 66.18 | 75.19 | 79.46 | 78.35 | 77.75 |
| **OPENBOOKQA** | Acc. | 60.64 | 72.49 | 80.32 | 40.76 | 73.30 | 85.94 | 66.06 | 65.66 |
| | Plau. | 0.49 | 0.59 | 0.67 | 0.66 | 0.73 | 0.74 | 0.55 | 0.55 |
| | Div. | 0.87 | 0.84 | 0.93 | 0.95 | 0.97 | 0.99 | 0.99 | 0.98 |
| | Cons. | 0.05 | 0.13 | 0.22 | 0.01 | 0.16 | 0.25 | 0.09 | 0.09 |
| | Avg. NRG | 62.29 | 68.00 | 75.33 | 63.07 | 75.33 | 80.36 | 68.64 | 68.29 |
| **NUMERSENSE** | Acc. | 26.13 | 48.24 | 61.81 | 17.59 | 36.18 | 74.37 | 55.28 | 54.27 |
| | Plau. | 0.51 | 0.65 | 0.72 | 0.62 | 0.68 | 0.76 | 0.63 | 0.63 |
| | Div. | 0.97 | 0.92 | 0.98 | 0.98 | 0.99 | 1.00 | 1.00 | 0.99 |
| | Cons. | 0.03 | 0.19 | 0.35 | 0.2 | 0.36 | 0.46 | 0.23 | 0.23 |
| | Avg. NRG | 56.41 | 66.19 | 74.83 | 59.40 | 67.80 | 80.84 | 69.95 | 69.44 |
| **QASC** | Acc. | 61.02 | 70.63 | 74.84 | 24.19 | 75.59 | 80.24 | 60.15 | 59.61 |
| | Plau. | 0.44 | 0.55 | 0.63 | 0.59 | 0.71 | 0.75 | 0.47 | 0.47 |
| | Div. | 0.78 | 0.63 | 0.89 | 0.74 | 0.98 | 0.97 | 0.99 | 0.99 |
| | Cons. | 0.23 | 0.32 | 0.37 | 0.10 | 0.31 | 0.38 | 0.19 | 0.19 |
| | Avg. NRG | 61.13 | 63.66 | 73.84 | 53.05 | 77.52 | 80.31 | 66.41 | 66.28 |

that the annotators have to work on. These tasks are hand curated that cater certain parts of the instruction – whether the annotators are reading the rationale correctly, or whether they are able to make appropriate connections between the rationale and the question. This weeds out a lot of annotators who do not understand the task or are cheating. We also weed out workers who are too 'fast' (completing the task in less than 5 seconds, which is indicative of potential slacking in the task). Third, we constantly monitor task responses and feedback provided to annotators about their task. We also collect feedback from them which we adapt in new versions of the task.

The final MTurk instructions and template that we land upon after the qualifications is shown in Figure 6 and 7

## K    LIMITATIONS

MARIO demonstrates promising improvements on the self-rationalization capability of small language models; we note that using MARIO on a small LM like T5-LARGE leads to considerable bridging of the gap between the quality of its rationales versus the quality of rationales generated by much larger language models. However, we note that the results are still very much dependent on the initially available data (since we heavily depend upon silver standard rationales generated by GPT-3 to give our model a warm start). Our method is also dependent upon the mathematical rewards that we use: as we discuss in Section 5, this is a very new and active area of research, and we as a research community are still figuring out what properties we need, and how to efficiently implement a good mathematical metric for them.

Table 11: The complete prompt of rationalization for STRATEGYQA. Demonstration examples are collected from Wei et al., 2022

**Q:** Do hamsters provide food for any animals?
Hamsters are prey animals. Prey animals provide food for predators.
**A:** So the answer is yes.

**Q:** Could Brooke Shields succeed at University of Pennsylvania?
Brooke Shields graduated from Princeton University. Princeton is ranked as the number 1 national college by US news. University of Pennsylvania is ranked as number 6 national college by US news. Princeton only admits around 6 percent of applicants as of 2018. University of Pennsylvania accepts around 9% of applicants as of 2018.
**A:** So the answer is yes.

**Q:** Yes or no: Hydrogen's atomic number squared exceeds number of Spice Girls?
Hydrogen is the first element and has an atomic number of one. To square a number, you multiply it by itself. The Spice Girls has five members.
**A:** So the answer is no.

**Q:** Yes or no: Is it common to see frost during some college commencements?
College commencement ceremonies often happen during the months of December, May, and sometimes June. Frost isn't uncommon to see during the month of December, as it is the winter.
**A:** So the answer is yes.

**Q:** Yes or no: Could a llama birth twice during War in Vietnam (1945-46)?
The War in Vietnam (1945-46) lasted around 6 months. The gestation period for a llama is 11 months.
**A:** So the answer is no.

**Q:** Yes or no: Would a pear sink in water?
The density of a raw pear is about 0.59 g/cm$\hat{3}$. The density of water is about 1 g/cm$\hat{3}$. Objects only sink if they are denser than the surrounding fluid.
**A:** So the answer is no.

Table 12: The complete prompt of rationalization for QUAREL. Demonstration examples are collected from Li et al., 2023a

---

**Q:** Mike was snowboarding on the snow and hit a piece of ice. He went much faster on the ice because ＿＿＿＿ is smoother.
(A) snow (B) ice
**A:** When something is smoother, it is easier to slide on. Thus, he could go faster on the ice because ice is smoother. So the answer is: (B)

**Q:** I could hear then boy that say close to me clear as day, however I could not hear the young lady sitting in the back of the room. Who am I able to hear louder
(A) Boy (B) Lady
**A:** When someone is close, it is easier to hear them. I also could not hear the young lady well. Thus, I am able to hear the boy louder. So the answer is: (A)

**Q:** I watched the snowflakes go from tiny specks in the sky to a nice size once they fell on my face. When did the snowflakes seem bigger?
(A) in the sky (B) on my face
**A:** When something is closer, it seems bigger. The snowflakes are closer when they are on my face. Thus, they seem bigger when they are on my face. So the answer is: (B)

**Q:** When Tammy tried to slide the glass mixing bowl down the marble counter top to her mom, it came to a dead stop when it reached the wooden cutting board. The bowl came to a stop because the wooden cutting board has
(A) more resistance or (B) less resistance
**A:** When something has more resistance, it is harder to slide. Thus, the bowl came to a stop because the wooden cutting board has more resistance. So the answer is: (A)

**Q:** Sarah walked through the city and saw a tourist attraction she wanted to visit. She had several blocks to go to get to it, and the attraction looked very small. As she got close to it though, it towered over her. This is because when she was close to it the attraction looked
(A) much bigger (B) much smaller
**A:** When something is closer, it looks bigger. Thus, the attraction looked much bigger when she was close to it. So the answer is: (A)

---

Table 13: The complete prompt of rationalization for OPENBOOKQA. Demonstration examples are collected from Wang et al., 2022

---

**Q:** The sun is responsible for
(a) puppies learning new tricks (b) children growing up and getting old (c) flowers wilting in a vase (d) plants sprouting, blooming and wilting
**A:** A plant requires sunlight for photosynthesis, which accumulates resources required for sprouting, blooming, and wilting. So the answer is: (d)

**Q:** When standing miles away from Mount Rushmore
(a) the mountains seem very close (b) the mountains are boring (c) the mountains look the same as from up close (d) the mountains seem smaller than in photographs
**A:** When an object is far away, it takes up less of your field of view, and so seems smaller than in the photographs. So the answer is: (d)

**Q:** When food is reduced in the stomach
(a) the mind needs time to digest (b) take a second to digest what I said (c) nutrients are being deconstructed (d) reader's digest is a body of works
**A:** The stomach is part of the digestive system. The breaking down of food into nutrients occurs in the digestive system. So the answer is: (c)

**Q:** Poison causes harm to which of the following?
(a) a Tree (b) a robot (c) a house (d) a car
**A:** A tree is a living thing. Poison causes harm to living things. So the answer is: (a)

**Q:** A magnet will stick to
(a) a belt buckle (b) a wooden table (c) a plastic cup (d) a paper plate
**A:** A belt buckle is made of metal. If a magnet is attracted to a metal, then that magnet will stick to that metal. So the answer is: (a)

**Q:** Deer are less safe in the woods because wolves
(a) have fur (b) howl (c) have claws (d) have tails
**A:** Claws are used by wolves to catch prey like deer. So the answer is: (c)

**Q:** An electric car causes
(a) more $CO_2$ emissions (b) equal $CO_2$ emissions (c) electric emissions (d) less $CO_2$ emissions
**A:** An electric car uses less gasoline than a regular car and thus causes less $CO_2$ emissions. So the answer is: (d)

---

Table 14: The complete prompt of rationalization for NumerSense. Demonstration examples are collected from Liu et al., 2022

---

**Q:** penguins have \<mask\> wings.
(A) no (B) zero (C) one (D) two (E) three (F) four (G) five (H) six (I) seven (J) eight (K) nine (L) ten
**A:** Birds have two wings. Penguin is a kind of bird. So the answer is (D).

**Q:** a parallelogram has \<mask\> sides.
(A) no (B) zero (C) one (D) two (E) three (F) four (G) five (H) six (I) seven (J) eight (K) nine (L) ten
**A:** A rectangular is a parallelogram. A square is a parallelogram. So the answer is (F).

**Q:** there are \<mask\> feet in a yard.
(A) no (B) zero (C) one (D) two (E) three (F) four (G) five (H) six (I) seven (J) eight (K) nine (L) ten
**A:** A yard is three feet. So the answer is (E).

**Q:** water can exist in \<mask\> states.
(A) no (B) zero (C) one (D) two (E) three (F) four (G) five (H) six (I) seven (J) eight (K) nine (L) ten
**A:** There states for matter are solid, liquid, and gas. So the answer is (E).

**Q:** a typical human being has \<mask\> limbs.
(A) no (B) zero (C) one (D) two (E) three (F) four (G) five (H) six (I) seven (J) eight (K) nine (L) ten
**A:** Human has two arms and two legs. So the answer is (F)

---

Table 15: The complete prompt of rationalization for QASC. Demonstration examples are collected from Wang et al., 2023a

**Q:** How do you reduce pollution? (A) igniting fuel and oxidiser (B) transportation technology (C) wasting (D) not recycling (E) burning fossil fuels (F) converting electricity to heat (G) water conservation (H) using less resources
**A:** Conserving resources has a positive impact on the environment. Use of resources affects the environment such as pollution. So the answer is: (H)

**Q:** what will move to another area if their habitat will no longer support them? (A) density (B) Birds (C) squids (D) humans (E) clouds (F) gravity (G) cows (H) Whales
**A:** If a habitat can no longer support animals then those animals will move to another area. Cows are social animals. So the answer is: (G)

**Q:** With the exception of allergies, what may cause a person to seek medical attention? (A) Contact with latex (B) a tree falling (C) Organs within the body. (D) Contact with baby chicks (E) prolactin release (F) Contact with peanut butter (G) hypothyroidism (H) Contact with microorganisms
**A:** Microorganisms can cause infections. Infections usually require medical treatment. So the answer is: (H)

**Q:** Lavender can induce (A) healing (B) energy (C) hormones (D) mutations (E) Heart rate (F) growth (G) symptoms (H) warmth
**A:** Healing requires rest. Lavender induces restful sleep. So the answer is: (A)

**Q:** what state is a liquid in when frozen? (A) vapor (B) dense (C) gas (D) cooled (E) steam (F) solid (G) boiling (H) cold
**A:** Freezing means changing from a liquid into a solid by reducing heat energy. Liquids freeze when they change to the solid state. So the answer is: (F)

**Q:** what unites to form a diploid zygote? (A) plant reproduction (B) Most plants (C) orchids (D) sperm and ova (E) salt and pepper (F) predator and prey (G) honeybees (H) diploids and zygotes
**A:** Gametes then unite in fertilization and form a diploid zygote. Collectively, the sperm and the ova are also referred to as gametes. So the answer is: (D)

**Q:** What absorbs all visible light? (A) apples (B) coal (C) Green (D) coral (E) skin (F) bamboo (G) glass (H) eyes
**A:** If an object is black then that object absorbs all visible light. Light grains are quartz, Black grains are coal. So the answer is: (B)

**Instructions (click to expand/collapse)**

# Main Instructions

In this HIT, you will read a Question and two Explanations that help you answer the question. These questions can be asking a certain fact or about real life commonsense. Later, you will also be shown the correct answer to this question. You will be answering a question about your preference between the two explanations, along with 3 follow-up questions.

# Your Task

Given the **question** and **two explanations**, you then need to answer the following question:

- **Explanation Preference:** *Which explanation do you prefer for the above question?* For explanations that are equivalent or the exact same, you can prefer both as well.

You will then be shown the **correct answer** to the above question. You then need to answer the following followup questions:

- **Support:** *Which of the explanation(s) support the correct answer?*
  - You need to determine if Explanation 1, 2, Both or None of them support the correct answer. By support, we mean that you can arrive at the correct answer via the explanation. See example 1.
- **Repeating:** *Which of the explanation(s) repeats over and over?*
  - Sometimes, explanations might repeat the same words and concepts again and again. See example 2.
- **Logicality:** *Which of the explanation(s) make logical sense?*
  - Here, we mean that you should check if the explanations make logical sense, **on their own**. Do not try to determine if the explanation is correct or supports the correct answer. Apply commonsense or general knowledge to answer this. See example 3.

Figure 6: **MTurk Instructions.** We show these instructions to turkers, along with a sample HIT, and more examples that contain special cases of each of the annotation questions.

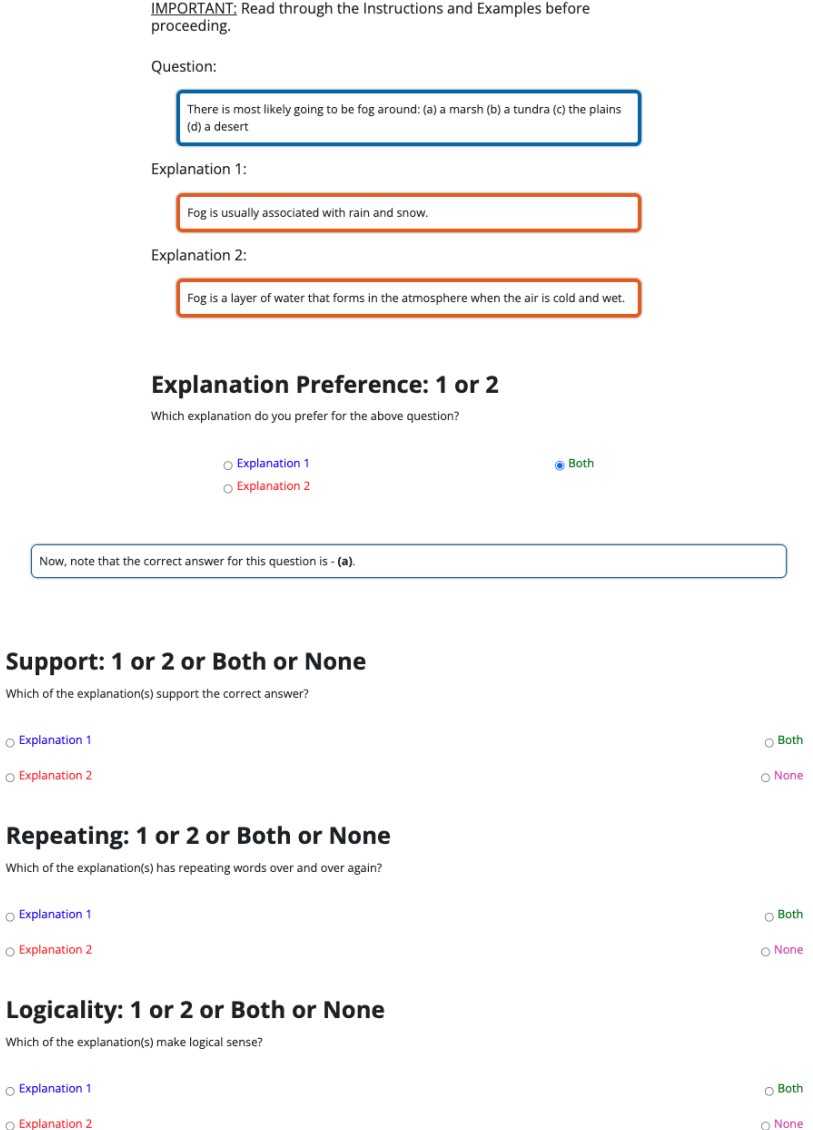

Figure 7: **MTurk Template.** Given a question and two explanations, we ask annotators to choose which explanation they prefer, followed by questions about their plausibility and consistency.

