# OpenReview forum: "Tailoring Self-Rationalizers with Multi-Reward Distillation"
_ICLR.cc/2024/Conference — ICLR 2024 poster_

### Official Review · Reviewer_CcB6 · 2023-10-15

**Soundness:** 3 good
**Presentation:** 3 good
**Contribution:** 3 good
**Rating:** 6
**Confidence:** 4

**Summary:**

The authors extend an existing LM control mechanism to work for controlling multiple attributes simultaneously, and following this mechanism they train T5-Large to do reasoning that is more plausible, non-repetitive, and consistent with the predicted answer. On some challenging QA tasks they get a small bump in accuracy, but a much bigger gain in human evaluations of the quality of the reasoning produced by the model to justify the answer.

**Strengths:**

--tackles a challenging task (making LMs output reasoning for their answers that is more reliable on a couple of different axes) which is highly topical

--impressive human eval results showing that their final rationales are much better than baselines'

--interesting discussion of reward hacking

**Weaknesses:**

--Methodologically, this multi-attribute control setup seems to be a straightforward extension of Quark, and the idea of finetuning an LM to use tags to provide control i think is pretty well-explored in other work such as [1], and doing it for multi-attribute with different tags is used as a baseline in [2]. i feel that your main contribution is not so much the new algorithm in a general sense, but rather the interesting application (with convincing human eval results) to the important and currently relevant task of making LM reasoning more reliable.

[1] Keskar, Nitish Shirish, et al. "Ctrl: A conditional transformer language model for controllable generation." arXiv preprint arXiv:1909.05858 (2019).

[2] Yang, Kevin, and Dan Klein. "FUDGE: Controlled text generation with future discriminators." arXiv preprint arXiv:2104.05218 (2021).

--some of the models you use to evaluate/filter the individual qualities, e.g. VERA, are way bigger than the model you're finetuning, which arguably gives you extra signal that the SFT baseline doesn't have? what was the data used for training those? on a related note, this raises some concerns that your method might not be scalable to larger LMs, unless we have e.g. an even larger version of VERA to provide supervision?

**Questions:**

--Is there any particular reason to think additive or classic might be better than the other in any particular setting? Otherwise, it kind of seems like you're just giving yourself "multiple tries" at your benchmark, in some sense.

--Is GPT3 = text-davinci-003 throughout?

--does adding the finetuning for consistency fix the problems in 5.2? or is this not exactly the same?

--the reward hacking discussion seems important and i'm glad you included it - there are a lot of potential hacks, e.g. a degenerate "rationale" could just be the answer itself or restating that in some way, right? do you get around this issue by starting with distillation so that the supervised rationales are initially reasonable, rather than doing e.g. some RL? i'm wondering how you would be able to go about this if you didn't have access to a much stronger model to distill from initially - it seems like so far you've only showed in a distillation-like setting, from GPT3 to a vastly smaller model.

--nit: maybe also cite [1]?

[1] Lightman, Hunter, et al. "Let's Verify Step by Step." arXiv preprint arXiv:2305.20050 (2023).

---

> ### Author Response · Authors · 2023-11-15
>
> Thank you for your positive review and very helpful comments and suggestions! We have added our response below as well as changes to the PDF based on your feedback.
>
> > Methodologically, this multi-attribute control setup seems to be a straightforward extension of Quark, and the idea of finetuning an LM to use tags to provide control i think is pretty well-explored in other work such as [1], and doing it for multi-attribute with different tags is used as a baseline in [2]. i feel that your main contribution is not so much the new algorithm in a general sense, but rather the interesting application (with convincing human eval results) to the important and currently relevant task of making LM reasoning more reliable.
>
> We partly agree with this! Yes, our task is tailored towards rationalization, which is coupled with accuracy and task performance. However, this is different from other controllable text generation tasks like creative text, poetry, machine translation, etc. However, we also note that adding multiple rewards for rationale generation was not a trivial addition, primarily because good quality rationales that increase a certain reward are not useful – they should be able to help improve the task accuracy, along with having high human preference. In Section 5.2, we can observe that LMs tend to overfit to a reward and hack their way to improving these metrics. We first need to select metrics that are actually important for rationalization, and at the same time, make sure that they are *compatible* in a way so that they can be used together. We hope this can help clarify the differences in our task setup and novelty behind our work.
>
> Thank you for linking the above works as well. As per this suggestion, we have added them in the subsection in our related work, where we discuss reward/multi-reward based generation methods
>
> > some of the models you use to evaluate/filter the individual qualities, e.g. VERA, are way bigger than the model you're finetuning, which arguably gives you extra signal that the SFT baseline doesn't have? what was the data used for training those? on a related note, this raises some concerns that your method might not be scalable to larger LMs, unless we have e.g. an even larger version of VERA to provide supervision?
>
> In this work, we focus majorly on how to use multiple rewards to improve the quality of rationales, both quantitatively and qualitatively. Therefore, how the rewards are actually designed are out of scope for this work. For example, for consistency, we use a t5-base reward model which is *smaller* than the t5-large we train. For plausibility, we use VERA (t5-11b) because the original work trained a t5-11b model on that property. In Section 5.2, we also talk about certain property metrics which depend on Web-API calls (such as factuality) which are practically infeasible - we experimented with zero-shot FLAN-T5 as a metric on factuality since we had to make a tradeoff between metric quality and inference time. Hence, say we want to train a much larger LM to self-rationalize, the reward models can still remain the same since they are a function of the property we want to measure and how good of a proxy they are for that metric, and not a function of the LM we are training.
>
> > Is there any particular reason to think additive or classic might be better than the other in any particular setting? Otherwise, it kind of seems like you're just giving yourself "multiple tries" at your benchmark, in some sense.
>
> The choice of MaRio Classic or Additive, and the choice of the order of the rewards in the multi-reward setting is a hyperparameter setting depending on the dataset we are training for. Sometimes, certain datasets require us to first optimize one reward, before adding the others one-by-one, whereas datasets might require all rewards to be provided from the very start for the best optimization; hence, we provide the framework for both such methodologies. (Additionally, other multi-reward techniques such as the product of rewards, or filtering of rewards have been added as baselines)
>
> (... continued below)

---

> > ### Author Response · Authors · 2023-11-15
> >
> > > Is GPT3 = text-davinci-003 throughout?
> >
> > Yes, we have edited the paper to reflect this when we are first referencing GPT-3.
> >
> > > Does adding the finetuning for consistency fix the problems in 5.2? or is this not exactly the same?
> >
> > No. In our prior experiments, we observed that adding consistency does not ensure that the rationale is complete in a sense, leading to accuracy drops.
> >
> > > the reward hacking discussion seems important and i'm glad you included it - there are a lot of potential hacks, e.g. a degenerate "rationale" could just be the answer itself or restating that in some way, right? do you get around this issue by starting with distillation so that the supervised rationales are initially reasonable, rather than doing e.g. some RL? i'm wondering how you would be able to go about this if you didn't have access to a much stronger model to distill from initially - it seems like so far you've only showed in a distillation-like setting, from GPT3 to a vastly smaller model.
> >
> > In current literature for rationalization, larger models like GPT-3 have good out of the box rationalization ability [1], which is why we use them for initial supervision. One alternative we can think of is to use a strong rationalizer out of the box, in the absence of a larger model for distillation – say Flan-T5-11B which can be a strong rationalizer with the correct set of demonstrations and prompts, without any other larger model for silver rationales. However, the larger our base model, the more time our reward-based updates would take. Quark in itself does sampling, filtering and updates for rewards, which we suspect would be infeasible for these large sized base models.
> >
> > [1] Wei et al. “Chain-of-Thought Prompting Elicits Reasoning in Large Language Models” (NeurIPS 2022)
> >
> > > nit: maybe also cite
> >
> > Thank you! We have updated this in our related work section and added the citation!

---

> > > ### Author Response · Authors · 2023-11-21
> > > **Friendly reminder to respond to author rebuttal**
> > >
> > > Dear Reviewer CcB6,
> > >
> > > Thank you again for your review! We are happy to hear that you appreciated our results (particularly human evaluation), the difficulty/challenge of our research question, and our discussions of reward hacking!
> > >
> > > Based on your thoughtful feedback, we have written a detailed rebuttal on the following points:
> > > * Controllable text generation with a particular focus on rationale generation (given the additional constraint that rationalization must lead to a good downstream task performance)
> > > * The choice of reward models
> > > * The design choice of MaRio Classic versus Additive
> > > * Missing citations about general controllable text generation
> > > * Extension to situations where distillation is not possible
> > >
> > > We would love to hear your thoughts about our rebuttal, including whether it sufficiently addresses your concerns and questions. If you believe that our rebuttal is satisfactory, it would be great if you could consider increasing your score. Any feedback is welcome and greatly appreciated!
> > >
> > > Sincerely,
> > >
> > > Paper-6506 Authors

---

> > > > ### Comment · Reviewer_CcB6 · 2023-11-22
> > > > **Thanks for the response**
> > > >
> > > > Thanks for addressing my questions. The methodology still seems quite similar to Quark to me, but I will maintain my score of 6 in view of the interesting application and empirical results.

---

### Official Review · Reviewer_7guo · 2023-10-29

**Soundness:** 3 good
**Presentation:** 3 good
**Contribution:** 3 good
**Rating:** 6
**Confidence:** 4

**Summary:**

This work studies the problem of generating plausible, consistent, and diverse rationales that also improve the downstream task performance while using small-scale LMs. The proposed method, called MARIO, is a multi-reward conditioned self-rationalization algorithm that optimizes multiple properties. Extensive evaluation show that produced rationales are plausible and improve performance.

Mario works a follow. First, it trains a small LM to self-rationalize, with the help of GPT-3. Then, it casts the problem into a multi-reward conditioned rationale generation problem. The key is to leverage a multi-reward setup, where the dimensions to optimize are plausibility, diversity, consistency, and task accuracy. Each reward is based on an automated metric. At the end, the propose method only extend Quark [Lu et al. 2022], which limits the novelty (the proposed method is 3 paragraphs).

The experiments focus on QA datasets. The two variants of Mario don't seem to significantly outperforms the baselines, Classical underperforms on StrategyQA and Mario on the others. However, the human evaluation show more promising results, showing that automated evaluations are not enough on their own. Other datasets than QA ones should be used to show the generalization of the proposed method. For example, beer, hotel, or amazon datasets. The analysis on reward hacking and optimizing solely on task performance is very interesting.

Overall the work is clear, well-written, and well-structured. It is a bit weird that the related work is put in the appendix. I would highly encourage the authors to move it back into the main paper. My concerns remain the novelty of the method - that seems to extend QUARK for multi-reward steup - and the lack of datasets that are not QA, especially when we are talking about rationalization.

POST-REBUTTAL:
Thank you for your answers; I am willing to increase my score.

**Strengths:**

- Good human evaluation and results
- Interesting method to make self-rationalization work with small LMs

**Weaknesses:**

- Other datasets than QA ones should be used to show the generalization of the proposed method.
- Small improvement in Table 2
- Limited novelty

**Questions:**

- How would you adapt your method for other rationalization tasks that are not QA?

---

> ### Author Response · Authors · 2023-11-15
>
> Thank you for your helpful comments and insightful questions! We are happy to hear that you appreciated our approach as well as human evaluations. Based on your feedback, we have updated the draft and provided the responses below.
>
> > Other datasets than QA ones should be used to show the generalization of the proposed method + How would you adapt your method for other rationalization tasks that are not QA?
>
> We essentially phrase the multi-choice QA task to be a classification task (Section 4.1), where we generate rationales for a given model prediction (class). Therefore, this can be easily extended to any classification task. We provide results on a range of QA tasks, each of which require varied types of natural language understanding (from factual to logical commonsense to numerical commonsense reasoning); we also show variety in the text structure of the options - StrategyQA requires a yes/no selection across all questions, OpenBookQA, QuaRel, QASC (** new) require a word/phrase to be selected, and Numersense (**new) requires the selection of a number. Furthermore, prior work in generating rationales and reasoning chains like Chain of Thought [1] and other subsequent work, also demonstrate this ability on QA tasks, which has shown to generalize on other tasks.
>
> [1] Wei et al. “Chain-of-Thought Prompting Elicits Reasoning in Large Language Models” (NeurIPS 2022)
>
> > Small improvement in Table 2 + Limited Novelty
>
> As we note in the paper as well, significant improvements over the SFT baseline denoted by *. As it can be seen, for most metrics, training using reward signals as done by MaRio leads to these numerical improvements. Furthermore, these numerical improvements translate to large improvements in human evaluations, where we see that across all of the datasets, annotators prefer MaRio generations more, w.r.t the baseline.
> Additionally, we noticed that adding multiple rewards for rationale generation was not a trivial addition. In Section 5.2, we can observe that LMs tend to overfit to a reward and hack their way to improving these metrics. We first need to select metrics that are actually important for rationalization, and at the same time, make sure that they are *compatible* in a way so that they can be used together. We hope this can help clarify the motivation and novelty behind our work.

---

> > ### Author Response · Authors · 2023-11-21
> > **Friendly reminder to respond to author rebuttal**
> >
> > Dear Reviewer 7guo,
> >
> > Thank you again for your review! We are happy to hear that you appreciated our experimental / human-evaluation-based results, and the interestingness of our method and our research question!
> >
> > Based on your thoughtful feedback, we have written a detailed rebuttal on the following points:
> > * The significance of improvements provided by MaRio in comparison to the baselines
> > * The novelty and importance of our contribution
> > * Our choice of datasets
> >
> > We would love to hear your thoughts about our rebuttal, including whether it sufficiently addresses your concerns and questions. If you believe that our rebuttal is satisfactory, it would be great if you could consider increasing your score. Any feedback is welcome and greatly appreciated!
> >
> > Sincerely,
> >
> > Paper-6506 Authors

---

> > > ### Author Response · Authors · 2023-11-23
> > >
> > > Thank you so much again for your feedback!
> > >
> > > We have updated the PDF based on your suggestions, and we have provided our rebuttal above, addressing all your questions and concerns!
> > >
> > > Today is the last day of the discussion period, and we would be grateful if you could check the rebuttal and let us know if it sufficiently addresses all your queries. If you believe that our rebuttal is satisfactory, it would be great if you could consider increasing your score. Any feedback is welcome and greatly appreciated!
> > >
> > > Sincerely,
> > >
> > > Paper-6506 Authors

---

### Official Review · Reviewer_pMad · 2023-10-30

**Soundness:** 4 excellent
**Presentation:** 3 good
**Contribution:** 3 good
**Rating:** 8
**Confidence:** 3

**Summary:**

In this paper, the authors presented MARIO, a method for training small LMs to generate rationales for question answering. The method is an extension to Quark. While Quark only allows optimizing towards one reward, MARIO allows learning from multiple-reward, allowing the LM to be trained with rewards for Plausibility, Consistency, Diversity and Task-correctness at the same time. The authors evaluated their method on 3 tasks: Strategy QA, QuaRel, and OpenBookQA, and the training rationales are sampled from InstructGPT outputs. The authors showed that their method outperformed baselines on similar-sized small LMs both on automated evaluation metrics and human preference, and can be comparable to some larger LMs on certain tasks.

**Strengths:**

1. The authors tackle an important problem: rationale generation for question answering on small LMs. It is known that rationalization and chain-of-thought can work better on very large language models, but fine-tuning small LMs to correctly rationalize in question-answering tasks has been very challenging.

2. The authors' proposed method allows learning towards multiple rewards, which can be very useful because often we want a model's generation to satisfy multiple desirable properties, and training towards a single property reward can often lead to complete loss of other desirable properties.

3. The authors comprehensively evaluated their method against several baselines on similar-sized LMs, and showed that their method is superior on 3 different QA tasks, both on automated metrics and human preference.

4. The presentation of the paper is generally quite clear. The figures and tables are very well made and they really help make the paper easier and better understood by the reader.

**Weaknesses:**

Overall this is a good paper. Below are a few weaknesses that prevented the paper from getting a "10":

1. The paper's main contribution is extending an existing method (Quark) from single-reward to multiple rewards. So while the results are nice and the extension is valuable, the contribution is not revolutionary.

2. While the description in text and Figure 2 are very helpful for readers to understand MARIO, the full picture of MARIO can still be a bit hard to grasp (especially to readers who are not already familiar with Quark). Maybe including an algorithm block for the entire training process will make the full picture a lot more clear.

**Questions:**

(1) Is there any way to rank/weight/balance different objectives in MARIO? (For example, if I found that the resulting model is weak in Consistency, is there a way to weigh Consistency reward a bit more in the training?)

---

> ### Author Response · Authors · 2023-11-15
>
> Thank you for your positive review and very helpful comments and suggestions! We are glad to hear that you appreciated our paper’s motivation, method and results. We have also added our response to the feedback and questions below.
>
> > The paper's main contribution is extending an existing method (Quark) from single-reward to multiple rewards. So while the results are nice and the extension is valuable, the contribution is not revolutionary.
>
> Thank you for recognizing our contribution and results! As far as novelty is concerned, our idea was that firstly, in prior rationalization literature, rationales were used as a means to an end to improve the accuracy of the system – as seen in Chain of Thought [1] and other following works. However, often these rationales act as a medium of communication between the LM and user – to improve trust [2], utility [3] as well as help improve human-LM collaboration [4]. This provided us with a strong motivation to improve the quality of rationales themselves, with an added side benefit of accuracy improvement. Secondly, we noticed that adding multiple rewards for rationale generation was not a trivial addition, as observed in Section 5.2, where we see that LMs tend to overfit to a reward and hack their way to improving these metrics. We first need to select metrics that are actually important for rationalization, and at the same time, make sure that they are compatible in a way so that they can be used together. Our work is an attempt to demystify the same as well. We hope this can help clarify the motivation and novelty behind our work.
>
> [1] Wei et al. “Chain-of-Thought Prompting Elicits Reasoning in Large Language Models” (NeurIPS 2022)
> [2] Chen et al. “Understanding the Role of Human Intuition on Reliance in Human-AI Decision-Making with Explanations” (CSCW 2023)
> [3] Joshi et al. “Are Machine Rationales (Not) Useful to Humans? Measuring and Improving Human Utility of Free-text Rationales” (ACL 2023)
> [4] Wiegreffe et al. “Reframing Human-AI Collaboration for Generating Free-Text Explanations” (NAACL 2022)
>
>
> > While the description in text and Figure 2 are very helpful for readers to understand MARIO, the full picture of MARIO can still be a bit hard to grasp (especially to readers who are not already familiar with Quark). Maybe including an algorithm block for the entire training process will make the full picture a lot more clear.
>
> Thank you for this suggestion! We have uploaded new figures to explain our motivation and method better! We also have a more detailed technical figure in the appendix (Figure 5) to explain Quark and MaRio side by side, and we have now included the actual objective function that we optimize for, during training. During camera ready, we will also add the algorithm block in the main text, while we adjust for space!
>
> > Is there any way to rank/weight/balance different objectives in MARIO? (For example, if I found that the resulting model is weak in Consistency, is there a way to weigh Consistency reward a bit more in the training?)
>
> That’s a great point actually. We started off with using these rewards individually (new single-reward optimization experiments added in Appendix G / Table 10), which showed that improving for one reward doesn’t necessitate improvements for other rewards. This motivated us to look into multi-reward approaches, leading to MaRio. Currently, we haven’t experimented with weighting (up or down) these metrics, and this is definitely an interesting future direction we are looking at!

---

> > ### Author Response · Authors · 2023-11-21
> > **Friendly reminder to respond to author rebuttal**
> >
> > Dear Reviewer pMad,
> >
> > Thank you again for your review! We are happy to hear that you appreciated our research question, our multi-reward algorithm MaRio, the competitiveness of our results, and our presentation of the work! Based on your thoughtful feedback, we have written a detailed rebuttal on the following points:
> >
> > * The novelty and importance of our contribution
> > * Extended algorithmic details of MaRio
> >
> > We would love to hear your thoughts about our rebuttal, including whether it sufficiently addresses your concerns and questions. If you believe that our rebuttal is satisfactory, it would be great if you could consider increasing your score. Any feedback is welcome and greatly appreciated!
> >
> > Sincerely,
> >
> > Paper-6506 Authors

---

> > > ### Author Response · Authors · 2023-11-23
> > >
> > > Thank you so much again for your feedback!
> > >
> > > We have updated the PDF based on your suggestions, and we have provided our rebuttal above, addressing all your questions and concerns!
> > >
> > > Today is the last day of the discussion period, and we would be grateful if you could check the rebuttal and let us know if it sufficiently addresses all your queries. If you believe that our rebuttal is satisfactory, it would be great if you could consider increasing your score. Any feedback is welcome and greatly appreciated!
> > >
> > > Sincerely,
> > >
> > > Paper-6506 Authors

---

> > ### Comment · Reviewer_pMad · 2023-11-30
> > **Rebuttal Acknowledgement**
> >
> > Thank you for your detailed response! My review remains strongly positive (8).

---

### Official Review · Reviewer_A7zX · 2023-11-01

**Soundness:** 3 good
**Presentation:** 2 fair
**Contribution:** 3 good
**Rating:** 6
**Confidence:** 3

**Summary:**

This paper introduces MARIO, a multi-reward approach that enhances the self-rationalization quality of small LMs and the performance of downstream tasks. The authors conducted experiments on three QA datasets and compared the results with several baselines, finding that MARIO can effectively train small LMs to generate rationales, which satisfy multiple distinct properties such as plausibility, diversity, and consistency, while also improving the performance of QA tasks. Additionally, human evaluation was carried out, confirming that the rationales generated by MARIO are more preferred by humans compared to the baselines. Furthermore, the paper discusses the importance of selecting appropriate rewards and preventing MULTI-REWARD HACKING.

**Strengths:**

The author presents an interesting and valuable research question, namely, how to enhance the self-rationalization quality of small LMs. Building upon the basis of quark, the paper effectively extends its application, utilizing multi-reward conditional generation to optimize both the rationale quality and the performance of downstream tasks. The article clearly explains the criteria for measuring three key aspects of a rationale's properties.

**Weaknesses:**

- The details of the MARIO algorithm are not adequately explained, such as how to determine the settings of control tokens, and the description of how to quantize samples under the quark framework is unclear (is it a comprehensive consideration of multiple attributes for ranking, or is it ranked based on a single attribute?).
- The description of the MARIO method is overly simplistic, and it lacks the necessary explanation of the thought process behind the development of this method.
- In relation to self-explaining rationalization, besides the generative rationales discussed in this paper, there is a series of extractive rationale works (such as Lei et al. 2016, Liu et al. 2023, and so on). Beyond the difference between generative and extractive approaches, the basic framework of these two types of work is very similar. Both require ensuring that the generated/extracted rationale is meaningful to humans while maintaining high performance in downstream tasks. Therefore, the related work section should also include this series of works.
   - Lei et al. 2016, Rationalizing Neural Predictions, EMNLP-2016
   - Liu et al. 2023, Decoupled Rationalization with Asymmetric Learning Rates, KDD-2023
- Although Figure 1 and Figure 2 contain a considerable amount of text, the information conveyed is limited.
- The experiment used multiple baselines, but in reality, it involves two baselines and their multi-reward forms of extension, lacking a comprehensive comparison with other works.

**Questions:**

Although the model utilizes the quark framework, it should clearly present the learning objectives in a multi-reward scenario. Since quark is a single-reward algorithm, its objective function under the extension of multi-reward goals is not intuitive. Could you provide a more detailed and clear explanation of MARIO and the corresponding multi-reward loss?

---

> ### Author Response · Authors · 2023-11-15
>
> Thank you for your positive review and very helpful comments and suggestions! We have added our response below as well as changes to the PDF based on your feedback.
> > The details of the MARIO algorithm are not adequately explained, such as how to determine the settings of control tokens, and the description of how to quantize samples under the quark framework is unclear (is it a comprehensive consideration of multiple attributes for ranking, or is it ranked based on a single attribute?).
>
> These details are present in the Appendix (due to space constraints). Appendix B (“Quark and MaRio”) addresses how Quark is used in MaRio. Appendix C (“Order of tokens”) is used to explain how we order our rewards for the multi-reward setup. Figure 5 reiterates over this diagrammatically. In Table 7, we also mention hyper-parameters used by the Quark and MaRio algorithms. We mention here that our method quantized scores into 5 bins for all the rationale reward, and 2 bins for the accuracy reward.
> Even for our multi-reward experiments, the generated data is quantized separately for each individual reward; the final set of control tokens for each data point is the concatenated set of individual control tokens. We have added a clearer description of this in Appendix B as well.
>
> > The description of the MARIO method is overly simplistic, and it lacks the necessary explanation of the thought process behind the development of this method.
>
> In section 5.2 and 5.3, we explicitly delve into the quirks of selecting good quality rewards, how these rewards actually translate into changes in the generated text, as well as a discussion about why we ended up adding task accuracy as another reward token in our design for MaRio. All of these serve as anchors for our design decisions for MaRio and highlight our thought process for the multi-reward setting. Our core algorithmic framework is based upon Quark itself, which is briefly explained in Appendix B. We hope the changes made in the appendix can help better clarify the method. If there are any other suggestions on how to explain our process better, we’d be more than happy to accommodate that into the draft!
>
> > In relation to self-explaining rationalization, besides the generative rationales discussed in this paper, there is a series of extractive rationale works (such as Lei et al. 2016, Liu et al. 2023, and so on). Beyond the difference between generative and extractive approaches, the basic framework of these two types of work is very similar. Both require ensuring that the generated/extracted rationale is meaningful to humans while maintaining high performance in downstream tasks. Therefore, the related work section should also include this series of works.
>
> Quark as a framework is explicitly designed for text generation, which is why our focus was primarily on free-text rationales. Additionally, we have also included a subsection on extractive rationales in our related work section based on your suggestion!
>
> > Although Figure 1 and Figure 2 contain a considerable amount of text, the information conveyed is limited.
>
> We acknowledge that Figures 1 and 2 were text dense. We have updated the PDF with new figures to explain our motivation and method better!
>
> > The experiment used multiple baselines, but in reality, it involves two baselines and their multi-reward forms of extension, lacking a comprehensive comparison with other works.
>
> We compare with (1) the standard baseline SFT [1] , (2) a closely related work StaR [2] and its multi-reward equivalent, (3) with instruction fine tuned and otherwise trained large LLMs [3]. This intends to cover methods that generate free-text rationales by fine-tuning or prompting approaches both. For reward-based rationalization, there are currently no other baselines; we attempt multiple prior methods of incorporating multiple rewards – e.g product of rewards. If we missed any other methods to compare to, kindly let us know and we’d be more than happy to add it to our comparison pool!
>
> [1] Marasović et al. “Few-Shot Self-Rationalization with Natural Language Prompts” (NAACL 2022)
> [2] Zelikman et al. “STaR: Bootstrapping Reasoning With Reasoning” (NeurIPS 2022)
> [3] Wei et al. “Chain-of-Thought Prompting Elicits Reasoning in Large Language Models” (NeurIPS 2022)
>
> > Although the model utilizes the quark framework, it should clearly present the learning objectives in a multi-reward scenario. Since quark is a single-reward algorithm, its objective function under the extension of multi-reward goals is not intuitive. Could you provide a more detailed and clear explanation of MARIO and the corresponding multi-reward loss?
>
> Based on the Quark algorithm, we have duly updated Appendix B (“Quark and MaRio”) with the objective function of MaRio. This is essentially the same as Quark, except that we have multiple reward tokens on which the generation is conditioned.

---

> > ### Author Response · Authors · 2023-11-21
> > **Friendly reminder to respond to author rebuttal**
> >
> > Dear Reviewer A7zX,
> >
> > Thank you again for your review! We are happy to hear that you appreciated our research question, and how our algorithm MaRio uses multi-reward conditional generation to optimize various rationale qualities as well as the downstream task performance!
> > Based on your thoughtful feedback, we have written a detailed rebuttal on the following points:
> > * Details behind the implementation and practical usage of the MaRio algorithm (ranking of the multiple rewards, settings of the control tokens etc.)
> > * The objective function of the multi-reward optimization in MaRio
> > * Making the motivation and algorithm figures more informative
> > * MaRio versus baselines
> > * Missing citations about extractive rationales
> >
> > We would love to hear your thoughts about our rebuttal, including whether it sufficiently addresses your concerns and questions. If you believe that our rebuttal is satisfactory, it would be great if you could consider increasing your score. Any feedback is welcome and greatly appreciated!
> >
> > Sincerely,
> >
> > Paper-6506 Authors

---

> > ### Comment · Reviewer_A7zX · 2023-11-22
> > **Acknowledgement**
> >
> > Thanks for your reply.

---

### Official Review · Reviewer_3RWN · 2023-11-04

**Soundness:** 3 good
**Presentation:** 3 good
**Contribution:** 2 fair
**Rating:** 6
**Confidence:** 3

**Summary:**

In this work, authors contribute to the task of rationale’s generations in question answering tasks. In previous work, rationalizers are at significant scale,  and ignored the semantics of the rationales themselves. In this work, authors invented the MARIO algorithm for much smaller scale LM's to generate higher quality rationales, with multiple rewards for different quality properties of generated text rationales. Besides, generations from the algorithms seem to be more perferred for human experts.

**Strengths:**

This paper is well-written. The novelty and contribution is clear to me. The authors try not to take advantage of the scalability of large language models and instead use a much smaller distilled version of GPT.  Furthermore, the rewards’ design is aiming at generating rationales with better semantic qualities rather than scoring better at the specific downstream task. I think this is a really good design philosophy for training algorithms.

**Weaknesses:**

I believe the improvement of two versions of Marios compared to baselines is not really significant, considering that all the baseline models have equal number of parameters with the Mario agent model. I’m wondering whether the extra efforts on training on multiple rewards are indeed worth it to improve the generations.

**Questions:**

In the paper, the authors mentioned that there is an initial supervision phase where GPT-3 provides the generation labels. I’m wondering what’s the relationship between this initial supervision process and the distillation of GPT-3? Is it before, after, or exactly the distillation process?

**Details Of Ethics Concerns:**

No ethics concerns.

---

> ### Author Response · Authors · 2023-11-15
>
> Thank you for your helpful comments and insightful questions! We are happy to hear that you appreciated our method’s novelty, constraints, as well as design methodology. Based on your feedback, we have updated the draft and provided the responses below.
>
> > I believe the improvement of two versions of Marios compared to baselines is not really significant, considering that all the baseline models have equal number of parameters with the Mario agent model. I’m wondering whether the extra efforts on training on multiple rewards are indeed worth it to improve the generations.
>
> While we do note that the baselines and MaRio have equal number of parameters, in Table 3 (previously, Table 2), we note significant improvements over the SFT baseline denoted by *. As it can be seen, for most metrics, training using reward signals as done by MaRio leads to these numerical improvements. Another reason why these extra efforts are helpful are seen in human evaluations, as shown by Figure 3. Across all of the datasets, annotators prefer MaRio generations more, while also noting improvements across these quality metrics, w.r.t the baseline. This can help us understand that small improvements numerically lead to large improvements in actual generations, thereby directly improving preference of these generations.
>
> > In the paper, the authors mentioned that there is an initial supervision phase where GPT-3 provides the generation labels. I’m wondering what’s the relationship between this initial supervision process and the distillation of GPT-3? Is it before, after, or exactly the distillation process?
>
> We use the rationales provided by GPT-3 in the following 3 places: (1) we use it to train the supervised fine-tuned model SFT, which serves as a baseline, (2) we use the aforementioned SFT as the reference model for the KL divergence loss in MaRio’s training process (as we mention in Appendix), and (3) we also add these silver rationales to the overall data pool of MaRio. We have updated the PDF of the paper to re-reflect mentions of these throughout the draft.

---

> > ### Author Response · Authors · 2023-11-21
> > **Friendly reminder to respond to author rebuttal**
> >
> > Dear Reviewer 3RWN,
> >
> > Thank you again for your review! We are happy to hear that you appreciated MaRio’s novelty, efficiency in terms of (much) smaller language models, and the fact that we focus on improving the rationales’ semantic qualities in addition to the downstream task! Based on your thoughtful feedback, we have written a detailed rebuttal on the following points:
> > * The significance of improvements provided by MaRio in comparison to the baselines
> > * The information about how and where all silver rationales from GPT-3 are used.
> >
> > We would love to hear your thoughts about our rebuttal, including whether it sufficiently addresses your concerns and questions. If you believe that our rebuttal is satisfactory, it would be great if you could consider increasing your score. Any feedback is welcome and greatly appreciated!
> >
> > Sincerely,
> >
> > Paper-6506 Authors

---

> > > ### Author Response · Authors · 2023-11-23
> > >
> > > Thank you so much again for your feedback!
> > >
> > > We have updated the PDF based on your suggestions, and we have provided our rebuttal above, addressing all your questions and concerns!
> > >
> > > Today is the last day of the discussion period, and we would be grateful if you could check the rebuttal and let us know if it sufficiently addresses all your queries. If you believe that our rebuttal is satisfactory, it would be great if you could consider increasing your score. Any feedback is welcome and greatly appreciated!
> > >
> > > Sincerely,
> > >
> > > Paper-6506 Authors

---

> > ### Comment · Reviewer_3RWN · 2023-11-23
> > **Thank you for the response**
> >
> > Thank you for addressing my questions. Now I understand better about the distillation process and where the empirical improvement is. I decide to increase my score to 6.

---

### Author Response · Authors · 2023-11-15

We thank all of the reviewers for their thoughtful feedback and recognition of our paper’s contributions! We have addressed individual reviewers’ comments in their replies, as well as updated the PDF with the changes suggested. All changes in the PDF are marked in *red* for ease of reference.

Additionally, we also made the following additions to the appendix, for aiding our efforts to better motivate and explain MaRio.
- In order to demonstrate additional reasoning abilities while rationalization, we’ve replicated our experiments on two additional datasets (Appendix F, Table 9, Figures 6 and 7).
- We have also added results corresponding to single-reward only optimizations in Appendix G to further demonstration MaRio’s holistic gains across *all* metrics.
- We have also added results with few-shot LLMs for our two new datasets in Table 11 (previously Table 8).

We would love to hear your thoughts about our rebuttal, including whether it sufficiently addresses your concerns and questions. If you believe that our rebuttal is satisfactory, it would be great if you could consider increasing your score. Any feedback is welcome and greatly appreciated!

---

### Author Response · Authors · 2023-11-23
**Summary message to reviewers and S/ACs.**

We once again thank all the reviewers for their thoughtful feedback and recognition of our paper’s contributions!

We were happy to find that all reviewers found the ***premise of our work interesting*** (our usage of small-sized LMs as rationalizers which improves the rationales’ semantic qualities in addition to the downstream task performance), and ***appreciated our findings***, especially the results demonstrated by human evaluations.

Further, we ***address the questions*** raised by the reviewers in our rebuttal:
* *Contributions*: Reviewers CcB6, 7guo and pMad pointed out our work’s novelty and primary contributions, which we have addressed in their respective rebuttals. In summary, we noted that our contribution is novel due to two primary reasons:
    * We stress that reward-conditioned generation of free-text rationales is a novel task setup that has not been explored earlier (earlier methods only focus on rationalization as a means to improve task performance, without investigating rationale quality itself)
    * Our framework deals with multi-reward rationalization, that by itself is a non-trivial extension due to models ‘hacking’ their way to ‘bad’ generations (Section 5.2)

* *Numerical Improvements*: Reviewers 7guo and 3RWN mentioned that our work had small numerical improvements.
    * We mentioned in our rebuttals how every numerical improvement has also been backed by significance testing.
    * We also conducted extensive human evaluations that demonstrated how these numerical improvements lead to tangible improvements for human preferences on the rationale generations, which in itself is not trivial to achieve.
    * Lastly, during the rebuttal, we also added results for two more datasets in Appendix F to further strengthen our improvements.

* *Paper Feedback*: Reviewers pMad and A7zX mentioned difficulty in reading Figures 1 and 2 which we have duly fixed in our updated PDF, with clearer figures. Reviewers A7zX and pMad requested for more clarification and details for our algorithm, MaRio, which we have also duly updated in the relevant sections in the Appendix (B and D).

Lastly, we also want to highlight the following ***additions*** we made to the appendix, for aiding our efforts to better motivate and explain MaRio.
* In order to demonstrate additional reasoning abilities while rationalization, we’ve replicated our experiments and analyses on two additional datasets, Numersense and QASC (Appendix F, Table 9, Figures 6 and 7).
* We have also added results corresponding to single-reward only optimizations in Appendix G to further demonstration MaRio’s holistic gains across *all* metrics.
* We have also added results with few-shot LLMs for our two new datasets in Table 11 (previously Table 8).

We have updated the PDF with the changes and additions described above. All changes in the PDF are marked in ***red*** for ease of reference.

We thank the reviewers again for their time and their helpful feedback.

We would love to hear your thoughts about our rebuttal, including whether it sufficiently addresses your concerns and questions. If you believe that our rebuttal is satisfactory, it would be great if you could consider increasing your score. Any feedback is welcome and greatly appreciated!

---

### Meta-Review · Area_Chair_rVZn · 2023-12-06

**Metareview:**

The paper proposes a method for training smaller language models to generate rationales in question-answering tasks. The proposed method utilizes multiple rewards to improve the quality of rationalization across different quality attributes. Extensive experiments on three datasets showcase the effectiveness of the proposed method. The reviewers agreed that the paper addresses an important problem in training smaller language models and that the work would be of broad interest to the community. However, the reviewers also raised several concerns and questions in their initial reviews. We want to thank the authors for their responses and active engagement during the discussion phase. The reviewers appreciated the responses, which helped in answering their key questions. The reviewers have an overall positive assessment of the paper, and there is a consensus for acceptance. The reviewers have provided detailed feedback, and we strongly encourage the authors to incorporate this feedback when preparing the final version of the paper.

**Justification For Why Not Higher Score:**

The paper could possibly be given a score of "Accept (spotlight)" based on score calibration with other accepted papers.

**Justification For Why Not Lower Score:**

"Accept (poster)" is justified as the reviewers have an overall positive assessment, and there is a consensus for acceptance.

---

### Decision · Program_Chairs · 2024-01-16

Accept (poster)